# Assessing the impact of non-pharmaceutical interventions (NPI) on the dynamics of COVID-19: A mathematical modelling study of the case of Ethiopia

Bedilu Alamirie Ejigu [1]*, Manalebish Debalike Asfaw [2], Lisa Cavalerie [3,4], Tilahun Abebaw[2], Mark Nanyingi[3,5], Matthew Baylis[3]

1 Department of Statistics, College of Natural and Computational Sciences, Addis Ababa University, Addis Ababa, Ethiopia, 2 Department of Mathematics, College of Natural and Computational Sciences, Addis Ababa University, Addis Ababa, Ethiopia, 3 Department of Livestock and One Health, Institute of Infection, Veterinary and Ecological Sciences, University of Liverpool, Liverpool, United Kingdom, 4 International Livestock Research Institute, Addis Ababa, Ethiopia, 5 Department of Epidemiology and Public Health, School of Public Health, University of Nairobi, Nairobi, Kenya

* bedilu.alamirie@aau.edu.et

**Data Availability Statement:** All dataset used in this manuscript are available from the EPHI

## Abstract

The World Health Organization (WHO) declared COVID-19 a pandemic on March 11, 2020 and by November 14, 2020 there were 53.3M confirmed cases and 1.3M reported deaths in the world. In the same period, Ethiopia reported 102K cases and 1.5K deaths. Effective public health preparedness and response to COVID-19 requires timely projections of the time and size of the peak of the outbreak. Currently, Ethiopia under the COVAX facility has begun vaccinating high risk populations but due to vaccine supply shortages and the absence of an effective treatment, the implementation of NPIs (non-pharmaceutical interventions), like hand washing, wearing face coverings or social distancing, still remain the most effective methods of controlling the pandemic as recommended by WHO. This study proposes a modified Susceptible Exposed Infected and Recovered (SEIR) model to predict the number of COVID-19 cases at different stages of the disease under the implementation of NPIs at different adherence levels in both urban and rural settings of Ethiopia. To estimate the number of cases and their peak time, 30 different scenarios were simulated. The results indicated that the peak time of the pandemic is different in urban and rural populations of Ethiopia. In the urban population, under moderate implementation of three NPIs the pandemic will be expected to reach its peak in December, 2020 with 147,972 cases, of which 18,100 are symptomatic and 957 will require admission to an Intensive Care Unit (ICU). Among the implemented NPIs, increasing the coverage of wearing masks by 10% could reduce the number of new cases on average by one-fifth in urban-populations. Varying the coverage of wearing masks in rural populations minimally reduces the number of cases. In conclusion, the models indicate that the projected number of hospital cases during the peak time is higher than the Ethiopian health system capacity. To contain symptomatic and ICU cases within the health

website www.ephi.gov.et/, and also included as a supplementary material.

**Funding:** This work was supported as a sandpit project of the One Health Regional Network for the Horn of Africa (HORN) project to Addis Ababa University, Ethiopia. The HORN project is funded by UK Research and Innovation (UKRI) and the Global Challenges Research Fund (GCRF) from the Growing Research Capability call awarded to the University of Liverpool, UK (project number BB/P027954/1). Funders had no role in the design, analysis and interpretation of the results.

**Competing interests:** The authors have declared that no competing interests exist.

system capacity, the government should pay attention to the strict implementation of the existing NPIs or impose additional public health measures.

# 1 Introduction

The novel Severe Acute Respiratory Syndrome Coronavirus 2 (SARS-CoV-2) was first discovered in Wuhan, China in December 2019 [1]. The associated disease, COVID-19 was declared as a pandemic by the World Health Organisation (WHO) on March 11, 2020. By November 14, 2020 over 53.5 million confirmed cases and 1.3 million deaths had been reported across the world [1]. In Africa, the burden and impacts of the pandemic are less compared with Europe and the USA. This may be attributed to the later arrival of the pandemic, low seeding rate, youthful demographics and possible prior immunity to coronavirus-like infections [2]. Nevertheless, the pandemic is accelerating in Africa and passed 2.0 million confirmed cases on November 13, 2020 according to the [3] situation report. In Ethiopia, the first imported COVID-19 case was detected on March 13, 2020 by the Ministry of Health in a Japanese traveller from Burkina Faso [4].

To control the spread of the disease, the Ethiopian government took immediate action, by closing schools, and banning sporting events and mass gatherings. It took 79 days to reach the first 1000 cases, but 10,000, 20,000, 40,000 and 80,000 cases were attained in shorter periods of 51, 16, 18 and 44 days respectively. By the middle of November, 2020 the cases had surpassed 100,000.

As the pandemic accelerates globally, many African countries, including Ethiopia, have been implementing different non-pharmaceutical interventions (NPIs) to contain the spread of the disease. The WHO is leading a global COVAX initiative for accelerated development of promising vaccine candidates that may be available in early 2021, and experimental therapies are also undergoing clinical trials for safety before licensing. Currently,the most effective means to control the spread of COVID-19 remains the implementation of different NPIs to break chains of transmission [5–9]. The transmission pathways of COVID-19 from person to person are: i) close contact through respiratory droplets, ii) direct contact with infected persons, and iii) contact with contaminated fomites (objects and surfaces) [10]. Public health measures are intended to diminish these transmission mechanisms.

Mathematical models have been previously used with greater success in understanding the transmission dynamics and control mechanisms of infectious diseases [11]. To understand the early transmission dynamics of COVID-19 under different scenarios, a number of mathematical models have been previously proposed [12–21]. These established epidemiological and mathematical models provide important insights for public health decision-makers to enforce different mitigation strategies in different countries. As COVID-19 continues to spread worldwide, most public health authorities are utilizing mathematical models for decision making on intervention guidelines. Many countries (for example the UK, China, Germany, USA, Morocco) revised their public health measures based on COVID-19 model predictions. We believe that, due to the differences in social interaction and life style of the urban and rural populations in Ethiopia, mathematical models developed in other countries may be inapplicable in unraveling the dynamics of disease in lower income settings. This study employs the use of mathematical models to simulate the spread and interruption of transmission of COVID-19 in Ethiopia.

Assessment of the effectiveness of the implemented and proposed intervention strategies for combating COVID-19 by predicting the number of new cases and deaths is a major challenge to the scientific community. [8, 22] estimated the impact of NPIs based on confirmed cases of COVID-19 in several countries. Similarly, [9] studied the impact of NPIs to reduce COVID-19 mortality and health care demand. To our knowledge, there has been very limited study of the impact of NPIs on the disease dynamics in the local context of Ethiopia. The only published study, by [23], focused on small, medium and large clusters of the population with social distancing, face masking and contact tracing implementation at different proportions. This study did not quantify the impact of hand washing measures on transmission dynamics. Furthermore, the study did not provide the estimated number of cases under different stages of the disease. [24] implements Auto-regressive Integrated Moving Average (ARIMA) modeling to project COVID-19 prevalence patterns in East African countries, mainly Ethiopia, Djibouti, Sudan and Somalia. However, the ARIMA model did not take into account the impact of NPIs with different adherence levels on the predicted number of cases.

Our study proposes a modified mathematical form of the classic SEIR model [11] and this modified model is used to compare the effect of different NPIs individually and in combination to mitigate COVID-19 in Ethiopia. This proposed model (1) has several advantages over previous works: i) it estimates time and size of the peak under the implementation of NPIs with different adherence levels in urban and rural population settings, ii) it differentiates asymptomatic and symptomatic infections which influences the number of ICU cases and deaths due to the disease, iii)it accounts for the effect of indirect transmission of the disease through contaminated environment, and iv) it provides the estimated impact of individual and combined public health measures on the dynamics of the disease.

## 2 Materials and methods

### 2.1 Representation of proposed mathematical model

In our modeling framework (Fig 1), the population is divided into different compartments according to the infection status of individuals: susceptible (S), exposed (E), asymptomatic infected ($I_a$), symptomatic infected ($I_s$), isolated at home or hospital with moderate health

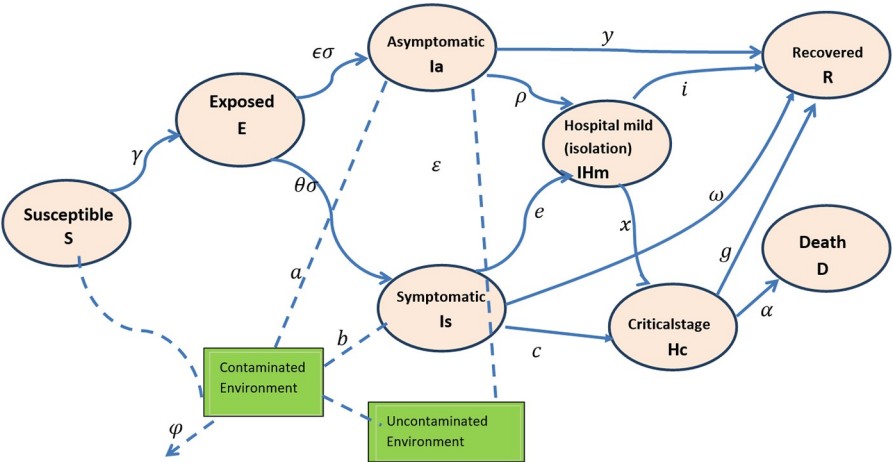

**Fig 1. Flow diagram of the mathematical model showing the transition of individuals in different compartments based on infection status (solid lines).** The dashed lines represent the potential shedding of the virus to the environment by individuals in the asymptomatic and symptomatic compartments as well as possible infection of the susceptible population from the contaminated environment at a rate dependent on the force of infection.

condition ($H_m$), hospitalized with severe health condition ($H_c$), recovered from the disease (R), and death (D) due to the disease. Contaminated environment is also a means of transmission for COVID-19 as the virus can stay up to several days on different surfaces [25]. To account for the impact of contaminated objects in virus transmission, an additional compartment for contaminated environment is included. Parameter descriptions are presented in Table 1. Assumed initial values were extracted from a number of key papers and reports of the situation in Ethiopia [4, 23, 26–28] as well as known epidemiological characteristics of COVID-19.

By assuming that the total population is $N = S + E + I_a + I_s + H_m + H_C + R + D$, the corresponding system of differential equations of our proposed model is given by:

$$\frac{dS}{dt} = \pi - \gamma S - \mu S$$

$$\frac{dE}{dt} = \gamma S - \sigma E - \mu E$$

$$\frac{dI_a}{dt} = (1-\theta)\sigma E - \rho I_a - \varepsilon I_a - y I_a - \mu I_a$$

$$\frac{dI_s}{dt} = \theta\sigma E + \varepsilon I_a - e I_s - \omega I_s - \alpha_1 I_s - \mu I_s \quad (1)$$

$$\frac{dIH_m}{dt} = \rho I_a + e I_s - x IH_m - i IH_m - \mu IH_m$$

$$\frac{dH_c}{dt} = x IH_m - g H_c - \alpha_2 H_c - \mu H_c$$

$$\frac{dR}{dt} = g H_c + i IH_m + y I_a + \omega I_s - \mu R$$

$$\frac{dD}{dt} = \alpha_1 I_s + \alpha_2 H_c$$

$$\frac{dE_{nv}}{dt} = a I_a + b I_s - \varphi E_{nv} \quad (2)$$

where the force of infection, $\gamma$, is obtained using the following formula:

$$\gamma = r_4 \beta_1 \left( \frac{(\eta_a I_a + I_s)}{N} \right) + r_4 \beta_2 \left( \frac{E_{nv}}{E_{nv} + K} \right). \quad (3)$$

In Eq (3), $\beta_1$ and $\beta_2$ are effective contact rates leading to COVID-19, $\eta_a$ is relative infectiousness per contact for asymptomatic patients, $r_4 = (1 - cFM)^*(1 - SD)^*(1 - HW)$, where SD represents the proportion of the population who practice physical distancing, HW represents the proportion of the population who implement hand washing, FM is coverage of wearing masks, c is efficacy of face mask wearing at reducing transmission, and K is the virus concentration in the environment that yields a 50% chance for a susceptible individual to catch the viral infection from that source [10]. The model accounts for a distinction between non-diagnosed individuals $I_a$ and $I_s$, who can readily spread the infection because they are not in isolation, and hospitalized individuals $H_m$ and $H_c$, who transmit the disease less thanks to isolation and complying with strict rules, either in hospital or at home. Details on the mathematical model system equilibria, stability of the system, sensitivity analyses, and well-posedness of the proposed mathematical model are provided as S1 File. Further, *Matlab-* codes used to get dynamic solutions of the systems of mathematical equations are presented in the S1 File. The dynamic solutions of the systems of equations in the considered model were obtained using *Matlab* [29]. In solving differential equations the Euler method was used. We believe it is better than other common methods because the Euler method is used in solving differential equations with

**Table 1. Description and values of initial model (1) parameters.** Parameter estimates were obtained from previous studies.

| Parameter | Description | Value | | Source |
|---|---|---|---|---|
| | | **Global-level** | **Ethiopia** | |
| $\gamma$ | Force of infection | | estimated | model 3 |
| $\beta_1$ | Rate of disease transmission directly from human | 0.81 | | [16] |
| $\beta_2$ | Rate of disease transmission from Contaminated environment | 0.4 | | [30, 31] |
| $\eta_a$ | Relative infectiousness of person in $I_a$ | 0.5 | | [17] |
| $\epsilon$ | Proportion of the exposed that goes to $I_a$ | 0.2 | | [16] |
| $\theta$ | Proportion of the exposed transfer to $I_s$ | 0.8 | | [31] |
| $\sigma$ | Rate at which exposed subjects are detected & move to $I$ | 0.2 | | [31] |
| $\alpha_2$ | Disease induced death rate | | 0.015 | [4] |
| a | Shedding rate from the asymptomatic Infectious class | 0.13 | | [16] |
| b | Shedding rate from the symptomatic infectious class | 0.10 | | [9] |
| $\varphi$ | Virus decays from the environment | 0.25 | | [30] |
| $\varepsilon$ | Rate of $I_a$ develop symp. and going to $I_s$ | 0.4 | | [31] |
| e | Rate of detected from $I_s$ going to $H_m$ | 0,8 | | [16, 32] |
| $\rho$ | Rate of detection & isolating or go to $H_m$ | 0.4 | | [9, 16] |
| y | Rate of recovery from $I_a$ | 0.2 | | [16, 32] |
| $\omega$ | Rate of recovery from $I_s$ | 0.4 | | [9, 16] |
| x | Rate of transfer from $H_m$ to $H_c$ | | 0.01 | [31] |
| $\alpha_1$ | Death Rate of transfer from $I_s$ | 0.1 | | [9, 16] |
| g | Rate of Recovery after getting critical | | 0.08 | [33] |
| i | Rate of Recovery after getting mild | | 0.60 | [33] |
| $\mu$ | Natural death rate | | 0.01 | [34, 35] |
| $\pi$ | Natural birth rate | | 0.03 | [36] |

known initial conditions. Further this method is also used to approximate the behavior of the equations involved in a certain range. To increase the accuracy of the method we used a time step of 0.01.

## 2.2 Projection of cases using the proposed model

The proposed modified SEIR model described in Section 2.1 is used to project the number of COVID-19 cases in Ethiopia. Projecting the number of active COVID-19 cases under the implementation of different NPIs with different adherence levels is important for policy makers to be able to mitigate against the disease. Projecting the number of asymptomatic and symptomatic people is important, as these have significant implications in the spread of the disease. Furthermore, knowing the expected number of people who may require treatment under intensive care (ICU cases) will greatly help to guide policy makers in optimizing the preparedness of healthcare workers and medical facilities. Due to the very limited number of available ventilators in Ethiopia (fewer than 1000 countrywide), it is important to estimate the number of severe cases that will require ICU hospitalization.

Based on the 2016 Ethiopian demographic and health survey, 27.4% and 7.8% of the urban and rural population wash hands using soap, respectively [37]. Thus, in the simulation, we assumed improved percentages of hygiene due to the awareness created by COVID-19 in both population settings. Due to differences in access to sanitation materials, lifestyle, cultural norms and other factors, adherence levels to the recommended NPIs are also different in the urban and rural populations. As a result, projection on the number of COVID-19 related cases is done separately for urban and rural populations of Ethiopia.

Table 2. Summary of considered adherence level coverage (%) to hygiene, wearing face mask and physical distance in urban and rural population of Ethiopia for numerical simulation.

| Residence | Hygiene | Physical distancing | Face mask |
|---|---|---|---|
| Urban | 30, 40 | 5, 10, 20, 30, 40 | 5, 15, 25 |
| Rural | 10, 20 | 5, 10, 20, 30, 40 | 0, 2, 5 |

## 2.3 Data

In this study, data on daily number of COVID-19 cases, cumulative number of deaths, and number of critical patients were extracted from the Ethiopian Public Health Institute website (www.ephi.gov.et/) and Ministry of Health official Twitter page(https://twitter.com/FMoHealth/) on a daily basis. We used initial parameter (Table 1) values describing the natural and clinical course of infection from published sources [1, 9, 10, 16]. The projection model considers daily number of cases in Ethiopia up to 210 days (October 08, 2020) since the first case of the disease was recorded.

In the rural parts of Ethiopia, physical distancing is a custom that is widely practiced. It greatly minimizes the spread of contagious diseases or those caused by contamination of the environment. Furthermore, due to low coverage of road accessibility (accessibility index = 22%), in our simulation we assumed 20% of the population did not travel frequently and therefore may be shielded from the high risk hence minimizing interactions or contact until the end of the pandemic [38]. Table 2 presents a summary of adherence levels used in the numerical simulation study. These values were selected based on expert opinion of how the dynamics may evolve under the implementation of NPIs with different adherence levels in the urban and rural populations. The coverage for the considered NPI's is supported by phone-based survey results [27, 28].

## 2.4 Considered Non-Pharmaceutical Interventions (NPIs)

Based on the current evidence about COVID-19, implementing different NPIs is essential for limiting the spread of the disease. The purpose of NPIs is to reduce the rate of transmission, thereby minimizing the size of the epidemic peak and delaying peak time, buying time for preparations in the healthcare system, and enabling the potential for vaccines and drugs to be developed, approved and obtained, [6–8, 39, 40]. The NPIs considered in this study are physical-distancing, wearing face-masks and hygiene measures (hand washing) (see Table 3). We assume that implementing NPIs alone or in combination affects the rate of contact between uninfected people and infected people or objects. As a result, transmission probability of the virus from infected individuals or objects to the susceptible population is reduced.

We evaluated the impact of the three NPIs, alone or in combination, on the time and size of the peak by varying the adherence level to each of them.

Table 3. Summary of considered intervention mechanisms and their effect on reducing the spread of COVID-19 based on existing literature.

| Intervention | Description | Reduction | References |
|---|---|---|---|
| Physical distancing | Creating ways to increase distance between people in settings where people commonly come into close contact with one another (i.e. schools, workplaces, events, spiritual centers, transport s, etc). | 35 times | [41] |
| Face-masking | Wearing face-masks protects susceptible individuals from acquisition of infection. If infected individuals wear face-masks, it reduces their ability to transmit the virus. | 30–70% | [16, 42] |
| Hygiene (hand washing) | Frequent, thorough hand washing with soap and water is effective at preventing the spread of lipid-enveloped viruses, such as SARS-CoV-2. | 40% | [43, 44] |

**Table 4. Summary of percentage change in $R_o$ under the implementations of different NPIs with different adherence levels in the urban and rural population of Ethiopia during the first 7 months of the pandemic.**

| | Number of days since the onset of the disease | | | | | | | | |
|---|---|---|---|---|---|---|---|---|---|
| | **2–60** | **61–90** | **91–105** | **106–120** | **121–140** | **141–160** | **161–172** | **173–184** | **185–210** |
| Urban | 44.08% | 12.00% | 57.83% | 62.41% | 46.83% | 51.42% | 44.75% | 58.75% | 58.75% |
| Rural | 44.13% | 48.72% | 56.96% | 62.45% | 46.88% | 50.55% | 51.46% | 63.37% | 60.62% |

## 3 Results: Projection of number of COVID-19 cases in Ethiopia

### 3.1 The basic reproductive number

The basic reproductive number ($R_o$) measures the rate of spread of SARS-CoV-2. It is the average number of secondary infections produced by a typical case of an infection in a population where everyone is susceptible If this number is greater than one, the disease will continue to spread to susceptible populations if no public health interventions are taken. In the absence of any NPIs, the estimated $R_o$ was 1.092 and 1.091 in rural and urban population, respectively.

Table 4 presents estimated percentage change in $R_o$ at different time points by taking into account enforced NPIs since the onset of the disease in urban and rural Ethiopia, respectively. Depending on the enforced NPI's by the government and its adherence level by the population, the percentage changes varies over time and higher in urban areas. Improving adherence to different NPIs greatly reduced $R_o$ in both urban and rural populations. As compared with the other estimated values of $R_o$, in the week after the holidays of Easter and Ramadan (61–90 days since the onset of COVID in Ethiopia), the percentage change in $R_o$ was lower because of the increased travelling and intense interaction in the population (Table 4).

Table 5 presents the summary of estimated $R_o$ under different NPIs. The results shows that, as compared with the urban population, the estimated value of $R_o$ is higher in rural areas.

### 3.2 Projection of all active cases

By fixing the adherence levels to hand-washing to 30% and 40% of the urban population based on aforementioned reasons, the projected number of active COVID-19 cases was obtained by varying the percentages of social distancing and face mask coverage. Fig 2 presents the projected number of active COVID-19 cases with varying adherence levels of NPIs.

At 15% face mask coverage and with 10% of the population implementing social distancing measures, a 10% increase in hygiene will decrease the number of cases by 200,000 and delay the peak time approximately by 55 days. For a given implementation of wearing a face mask and hygiene, improving the percentage of social distancing by 10% will reduce the number of

**Table 5. Summary of $R_o$ under the implementation of different NPIs with different adherence levels in urban and rural populations of Ethiopia.** The prevalence of hygiene measures (hand washing) is assumed to be 30% and 20% in urban and rural population of Ethiopia after the first seven months.

| Physical-distancing (%) | Urban | | | Rural | | |
|---|---|---|---|---|---|---|
| | Face mask coverage (%) | | | Face mask coverage(%) | | |
| | **5** | **15** | **25** | **0** | **2** | **5** |
| 5 | 0.56 | 0.41 | 0.45 | 0.83 | 0.79 | 0.75 |
| 10 | 0.67 | 0.68 | 0.66 | 0.70 | 0.69 | 0.66 |
| 20 | 0.45 | 0.64 | 0.62 | 0.61 | 0.6 | 0.56 |
| 30 | 0.59 | 0.57 | 0.55 | 0.53 | 0.52 | 0.45 |
| 40 | 0.52 | 0.50 | 0.48 | 0.83 | 0.82 | 0.76 |

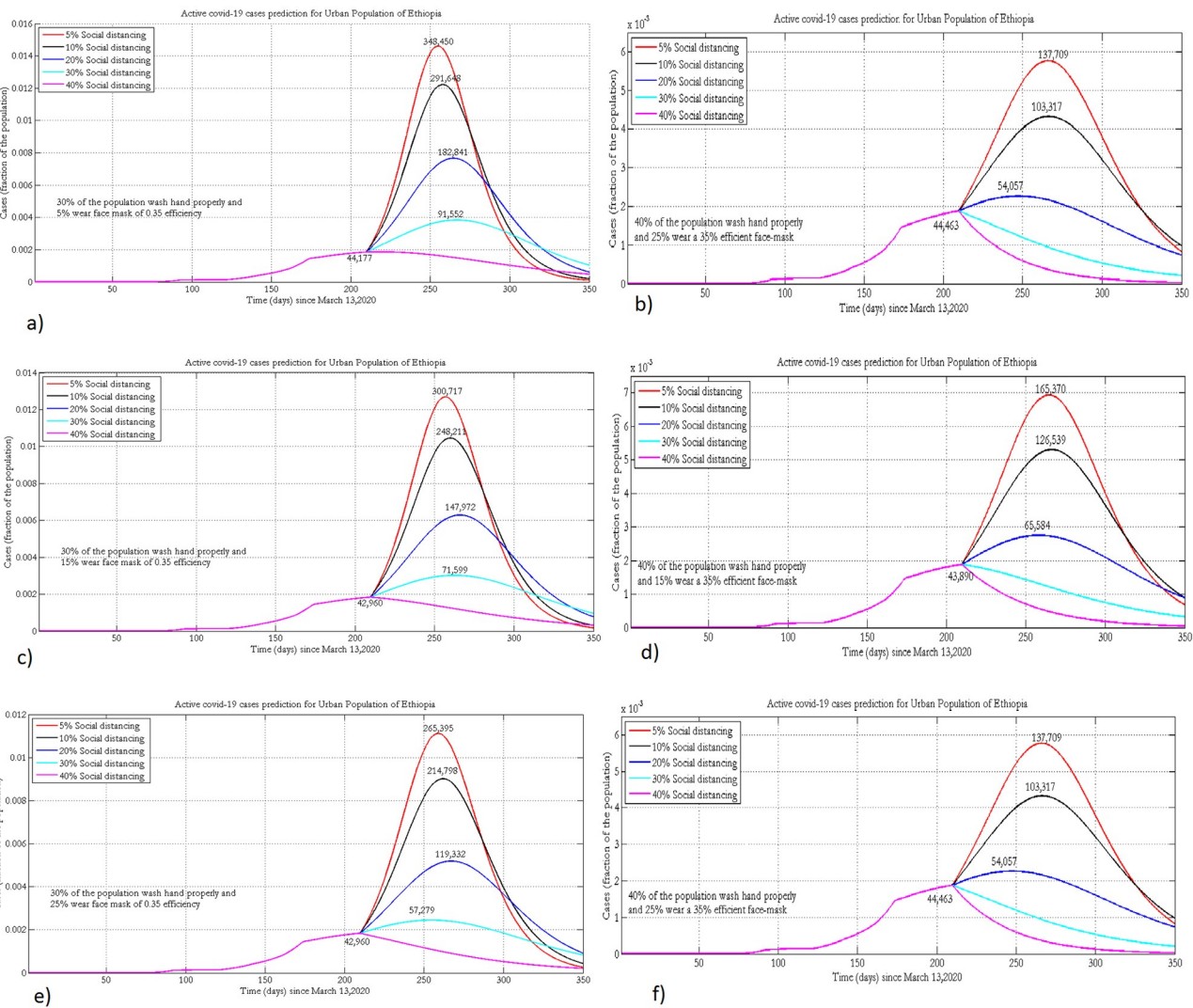

**Fig 2. Projection of active COVID-19 cases in the urban population of Ethiopia where 30% or 40% practice hand washing and 5% (a, b), 15% (c, d) or 25% (e, f) of the population wear face masks, with 35% efficacy.**

new COVID-19 cases by one-fifth. Increasing the number of people who wear face masks by 10% will prevent on average 25,000 people from being infected by the disease (Fig 2).

Fig 3 depicts the projected number of active cases under the implementations of varying physical distancing measures and wearing of face masks in the rural population setting. Implementing hygiene measures, physical distancing, and wearing face masks at 20%, 40%, and 5%, respectively, could shift the peak time to 2021. The projected number of active cases during the peak time could reach around 360,000 if 5%, 40%, and 20% of the population wear masks, keep physical distancing and hand wash with soap, respectively. As social distancing is already a custom due to the life style in the rural population of Ethiopia, improving hygiene by 20% could help to decrease the number of cases by 2–3 fold.

Keeping adherence to face mask wearing and physical distancing constant, a 5% increase in hand washing will decrease the number of cases by 200, 000 and shift the peak time to the future by nearly one month (Fig 3). For a given implementation of face mask wearing and

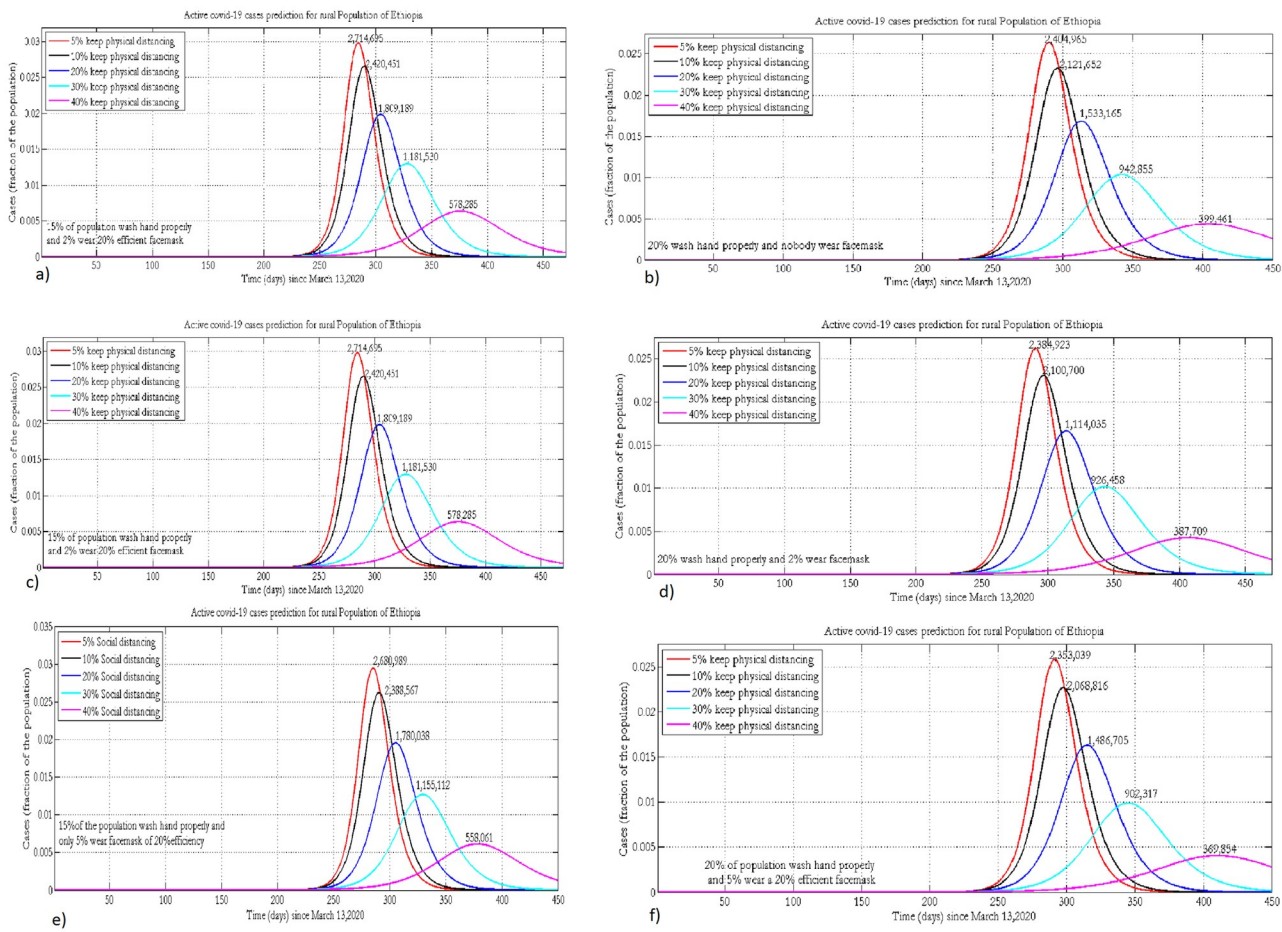

**Fig 3. Projection of active COVID-19 cases in the rural population of Ethiopia where 15% or 20% practice hand washing and 0% (a, b), 2% (c, d) or 5% (e, f) of the population wear face masks, with 20% efficacy.**

hygiene, improving the percentage of physical distancing by 10% will reduce the number of new COVID-19 cases by one-fourth.

### 3.3 Projection of symptomatic and asymptomatic cases

The following projected number of symptomatic (broken line) and asymptomatic (solid line) cases were simulated by assuming 30% and 40% of the urban population wash hands with soap, with different face mask coverage (5%, 15%and 25%) with 35% mask efficacy.

When 40% of the urban population wash hands with soap and, 25% wear face masks, improving the implementation of physical distancing greatly reduces both symptomatic and asymptomatic infections. By increasing the percentage of physical distancing in the population from 10% to 20%, the peak size will decrease by half for both infection compartments (Fig 4).

For a given coverage of face mask wearing and implementing physical distancing, improving hygiene by 10%, greatly reduces the number of active cases and delays the peak time of the pandemic. For both hygiene scenarios, for a given level of face mask wearing, increasing adherence to physical distancing profoundly reduces the number of symptomatic and asymptomatic cases. A maximum number of asymptomatic infections will be observed if there is poor

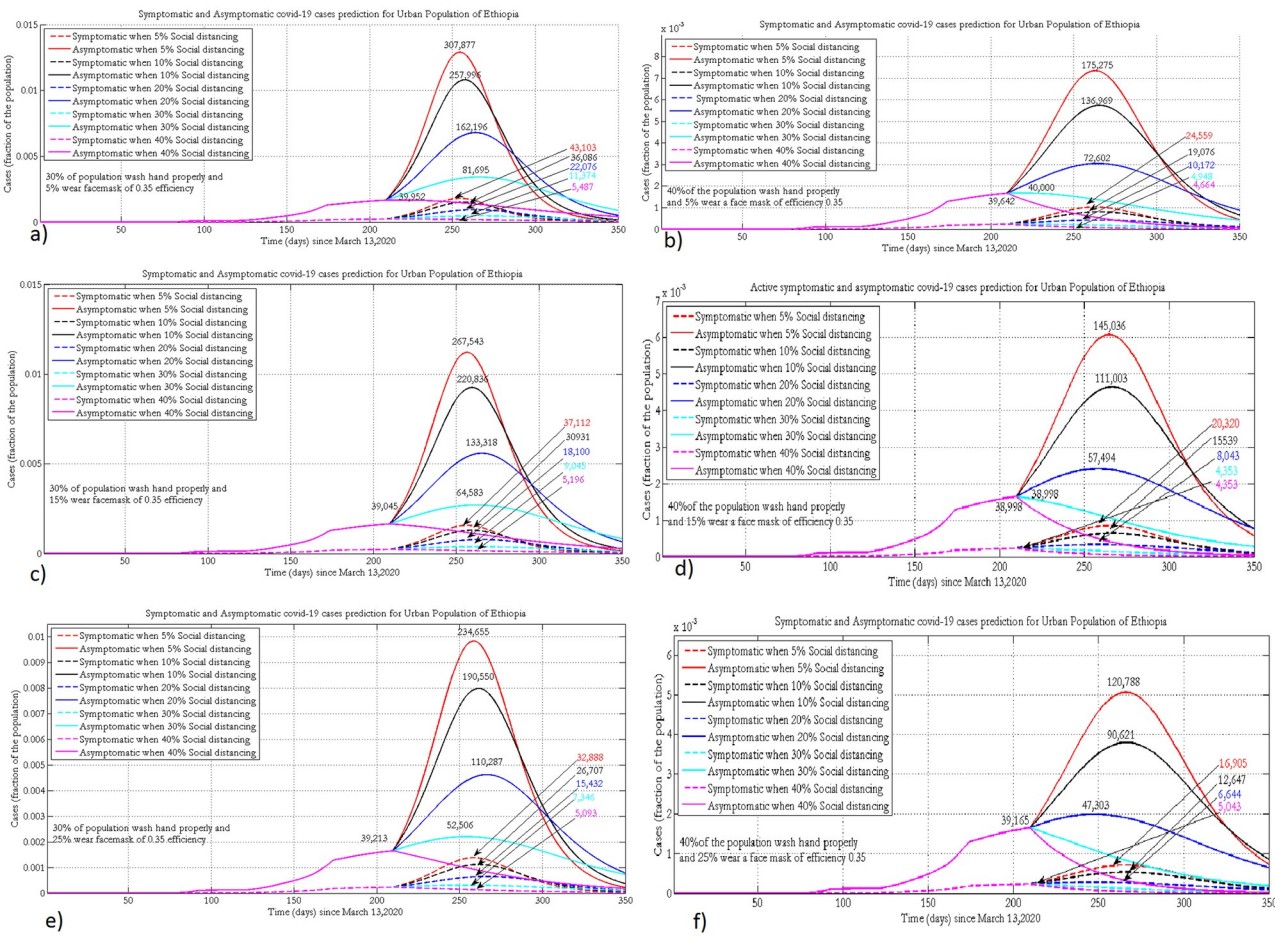

**Fig 4. Projection of symptomatic and asymptomatic infectious cases in urban populations of Ethiopia when 30% or 40% practice hand washing and 5% (a, b), 15% (c, d) or 25% (e, f) of the population wear face masks, with 35% efficacy.**

implementation of physical distancing and face mask wearing. Regardless of other NPIs, COVID-19 will be suppressed if 40% of the population implements physical distancing properly.

Fig 5 presents the projected number of symptomatic (broken line) and asymptomatic (solid line) cases in rural Ethiopia by varying the level of face mask wearing and physical distancing while fixing hygiene practice in the community at 15% and 20%. Increasing proper hand washing behaviour from 15% to 20% could shift the peak by around 2 months. Implementing physical distancing, hygiene, and face mask wearing at 40%, 20% and 5% levels, respectively, helps to control the pandemic to within the capacity of the Ethiopian health system and shifts the peak forward to 2021. At a given level of implementation of hygiene and wearing of face masks, increasing physical distancing by 10% shifts the peak time by around 40 days (Fig 5).

### 3.4 Projection of critical cases

Fig 6 presents the projected number of ICU cases by keeping the hygiene coverage constant (at 30%and 40%) and varying adherence to physical distancing and wearing of face masks at different levels in the urban population of Ethiopia.

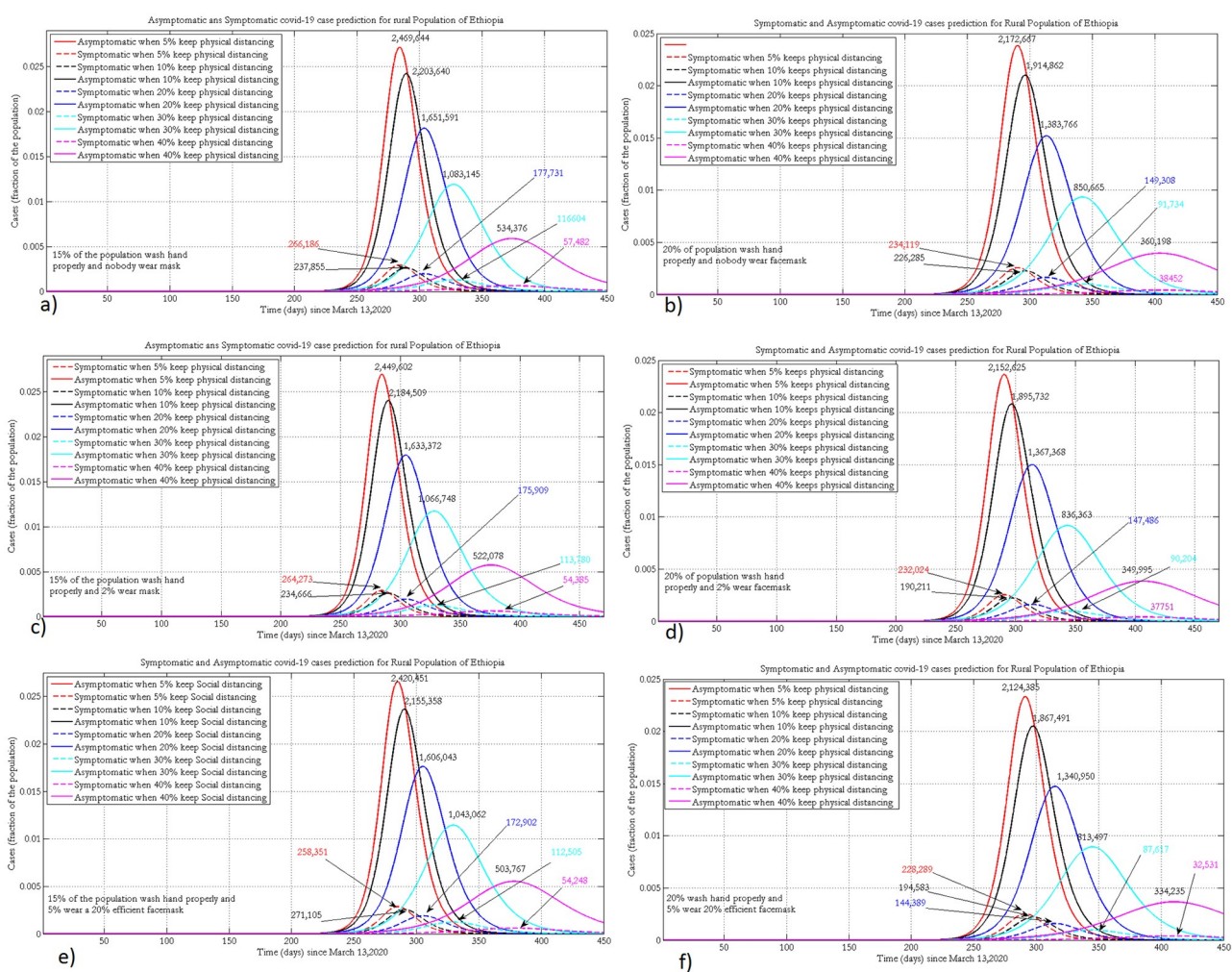

**Fig 5. Projection of symptomatic and asymptomatic infectious cases in the rural population of Ethiopia when 15% or 20% practice hand washing and 0% (a, b), 2% (c, d) or 5% (e, f) of the population wear face masks, with 20% efficacy.**

If 15% of the population wears masks and 30% implement physical distancing, improving hygiene by 10% could reduce the number of ICU cases from 463 to 280 (Fig 6). Proper implementation of wearing face masks and hand washing could flatten the curve. However, to bring critical cases within the Ethiopian health care system, people should also practice social distancing. To completely suppress the spread of the disease in all urban populations of Ethiopia, there has to be strict implementation of NPIs, where 40% have to practice hand washing, 40% physical distancing and 25% wearing masks.

In the rural population the projected number of critical cases could exceed the available number of ventilators in the Ethiopian healthcare system (Fig 7). Since the time of the peak in rural populations is different from that of urban populations, the healthcare system can make preparations for the transport system to bring these critical cases to heath facilities in urban centers. The two extreme scenarios show that improving hygiene by 5% and physical distancing by 40% could reduce the number of critical cases from 2,361 to 1,569 (assuming 5% wear masks) at peak time. Strict implementation of the three NPIs will shift the peak time forward by three months (Fig 7).

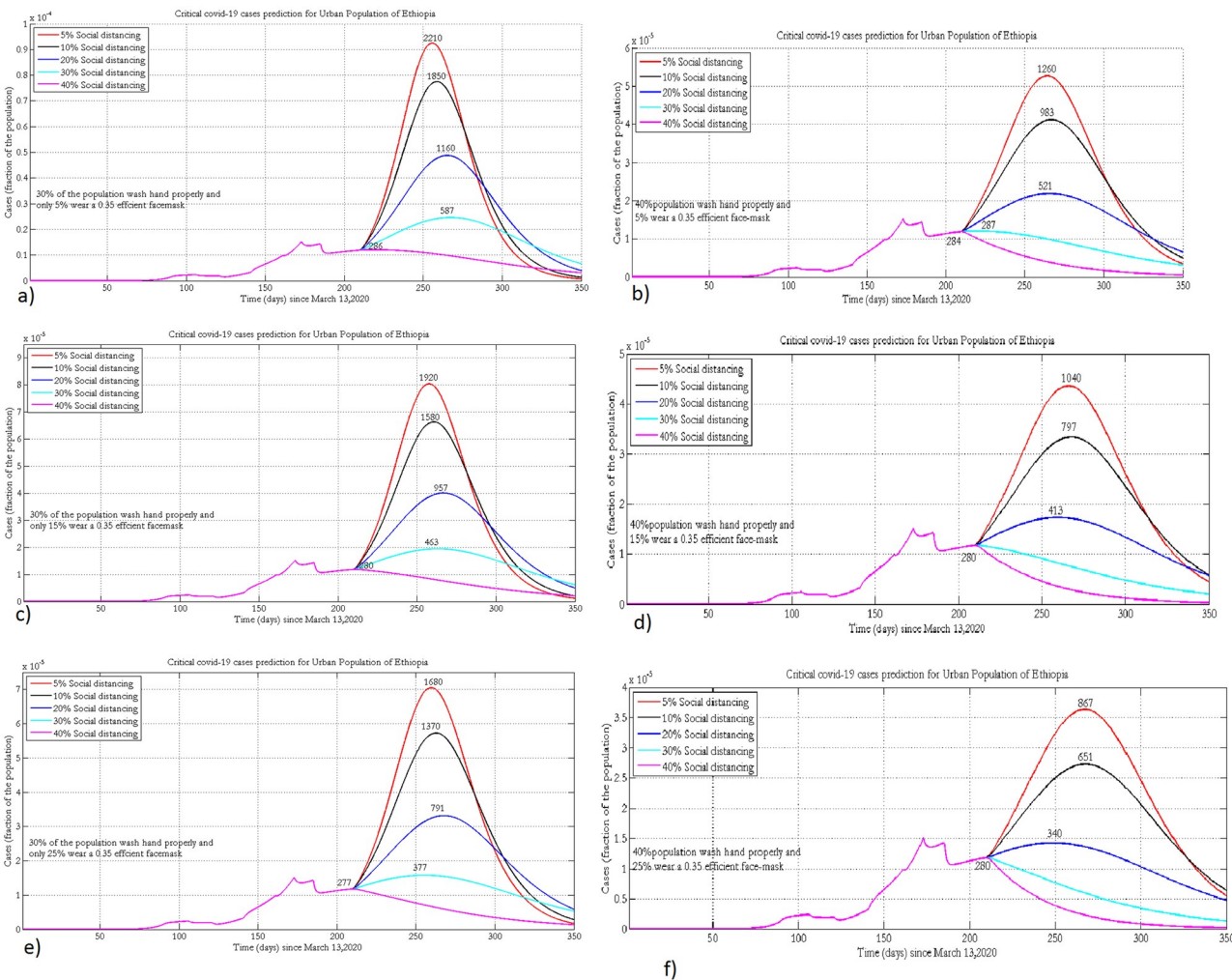

**Fig 6. Projected number of the urban population who will need intensive care when 30% or 40% practice hand washing and 5% (a, b), 15% (c, d) or 25% (e, f) of the population wear face masks, with 35% efficacy.**

## 3.5 Projection of cumulative deaths due to COVID-19

Fig 8 presents the projected number of deaths due to COVID-19 under different scenarios in urban populations of Ethiopia.

The number of deaths due to COVID-19 could be reduced if more than 40% of the populations practice physical distancing. The projected number of cumulative deaths under the implementation of different adherence levels of physical distancing and face mask wearing at fixed (15% and 20%) levels of hygiene are presented in Fig 9.

Practicing recommended NPIs properly (40% physical distancing 5% face mask wearing and 20% hand washing) could greatly reduce the number of deaths due to COVID-19. Tables 6 and 7 present summaries of the projected number of active COVID-19 and ICU cases under different adherence levels of hand washing, mask wearing and physical-distancing in the urban and rural populations of Ethiopia, respectively.

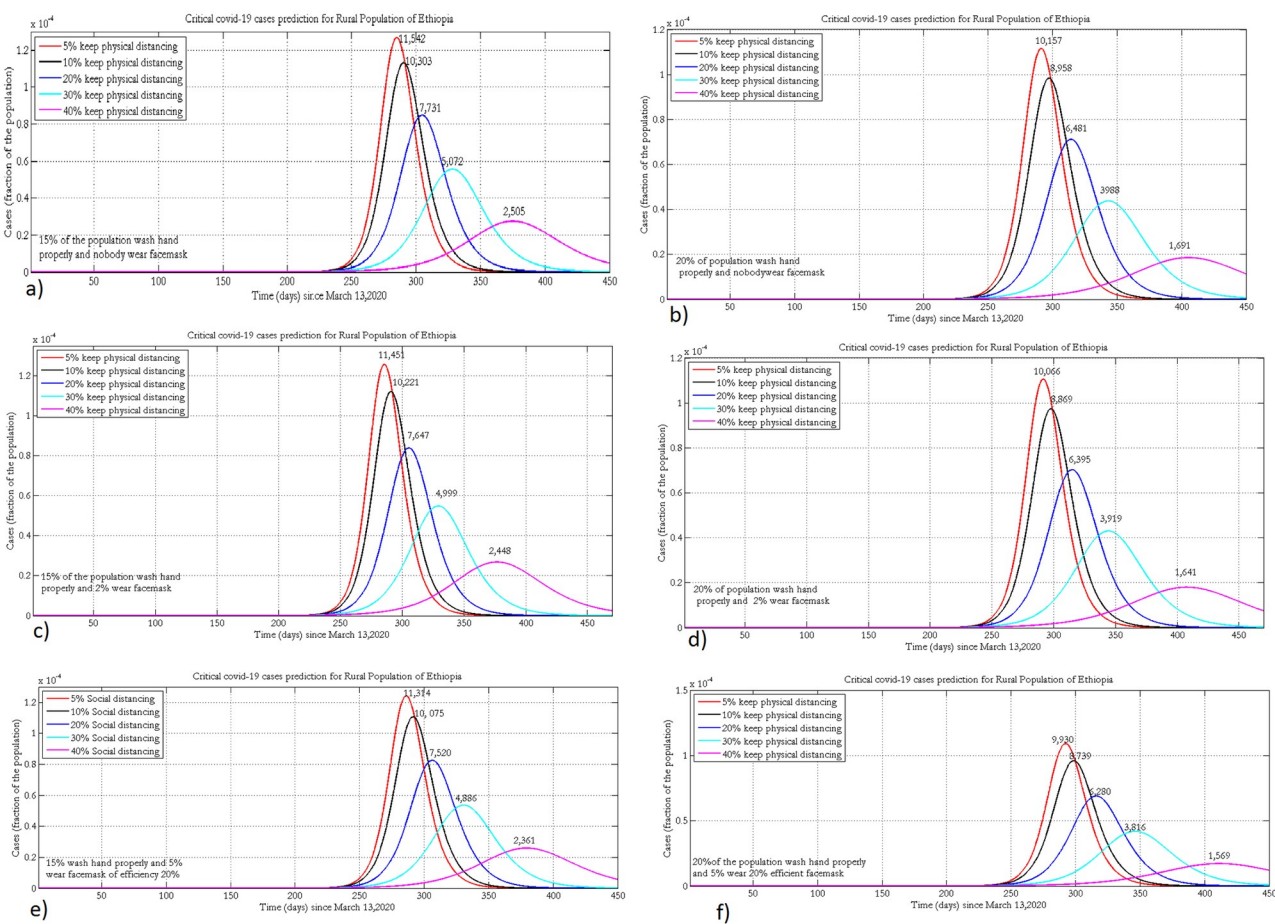

**Fig 7. Projection of intensive care or critical cases in the rural population of Ethiopia when 15% or 20% practice hand washing and 0% (a, b), 2% (c, d) or 5% (e, f) of the population wear face masks, with 20% efficacy.**

## 3.6 Sensitivity analysis

Fig 10 shows that the model is particularly sensitive to social distancing (SD), hand washing (HW) and the direct human to human (H2H) disease transmission rate ($\beta_1$). Hand washing and social/physical distancing were inversely proportional to $R_o$, while $\beta_1$ was directly proportional to $R_o$.

Parameters with relatively large Partial Rank Correlation Coefficient (PRCC) values ($>0.5$ or $< -0.5$) as well as corresponding small p values ($p < 0.05$) are considered to have the most significant influence on outcome of disease. Parameters whose PRCC value are close to $+1$ or $-1$ most strongly influence the model. A negative sign for PRCC indicates an inverse relationship of the parameter with $R_o$.

To explore the interdependence of parameters, pairwise comparisons were carried out. The processes underlying physical distancing, hand washing and the proportion of the population who become symptomatic, $\theta$, have the greatest potential of containing the epidemic if increased, whereas processes described by $\beta_1$ and $\epsilon$ and $\eta_a$ have the greatest potential of making the epidemic worse when increased (Fig 10).

In this respect, increasing physical distancing directly reduces $\beta_1$ as this lowers the likelihood of a susceptible individual making contact with a potentially infectious individual. In addition, practicing good hygiene (such as regularly washing hands, using sanitizers to

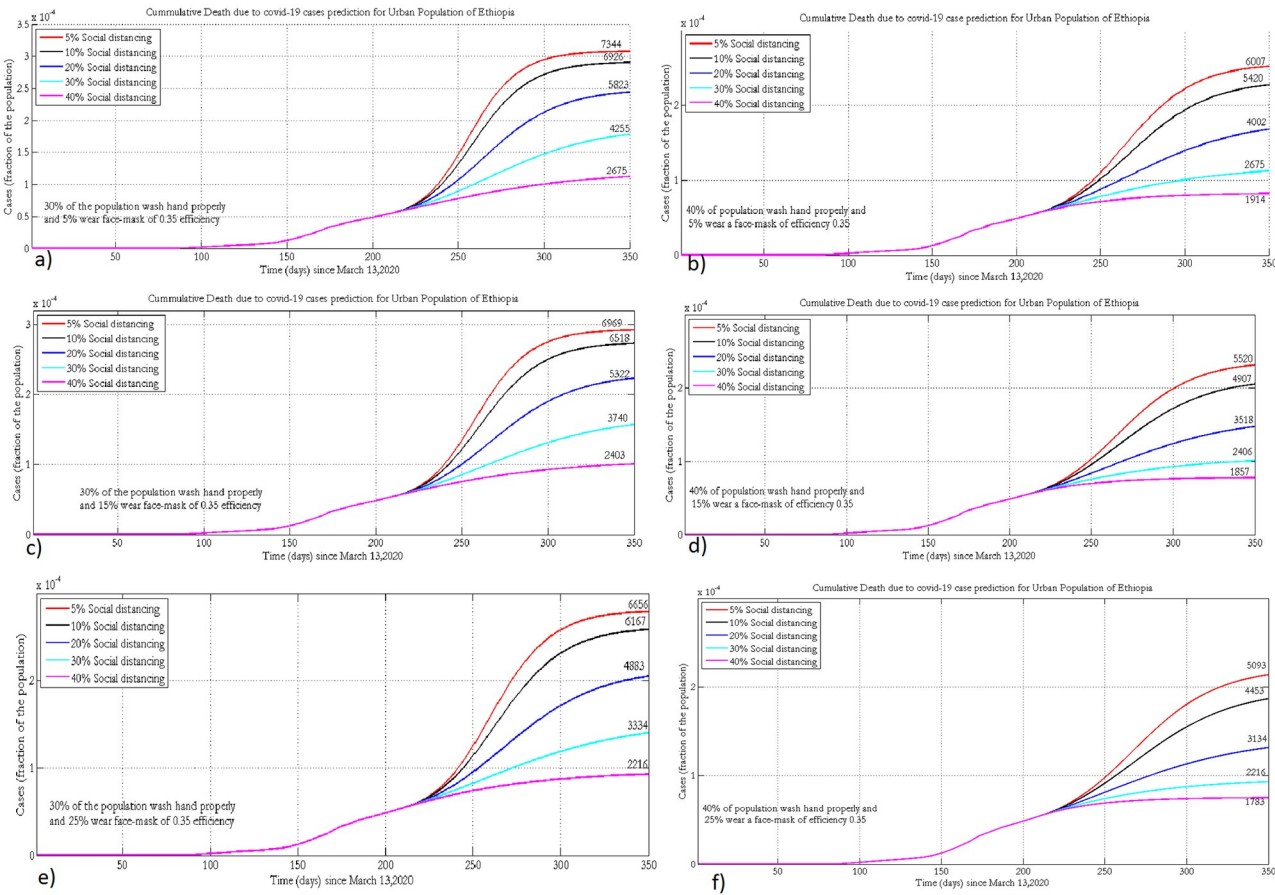

**Fig 8. Projection of deaths under the implementation of different NPIs when 30% or 40% practice hand washing and 5% (a, b), 15% (c, d) or 25% (e, f) of the population wear face masks, with 35% efficacy.**

disinfect the infected environment, ventilation of rooms and avoiding touching the T-zones of the face) is associated with lowering the chance of contracting the virus from contaminated surfaces. Anything contrary to the above increases the likelihood of getting the infection through the two aforementioned routes. Moreover, improving hand washing practices will reduce viral contamination of the environment by infected individuals.

On the other hand, the processes underlying the parameters with negative PRCC have a potential to contain the number of cases when enhanced.

To ascertain whether the process described by those parameters are different or not, a pair-wise comparison to significant parameters was undertaken. Table 8 presents the computed p-values for different pairs of significant parameters by accounting for the false discovery rate (FDR). The major question posed at this point is: are the pairs of significant parameters different after FDR adjustment? Based on the FDR adjusted p-values in Table 8, the compared pairs of parameters are found to be different if their p-value is less than 0.05 and not different otherwise.

Table 8 presents a summary of the compared parameters, where "TRUE" indicates compared parameters are significantly different, and "FALSE" otherwise. The results show that more sensitive parameters are also significantly different (see Table 8) except for the pair rate of recovery of asymptomatic infectious individuals and-appropriate use of face masking $y - FM$, recovery rate of symptomatic infectious individual and-rate of going to isolation $w - e$,

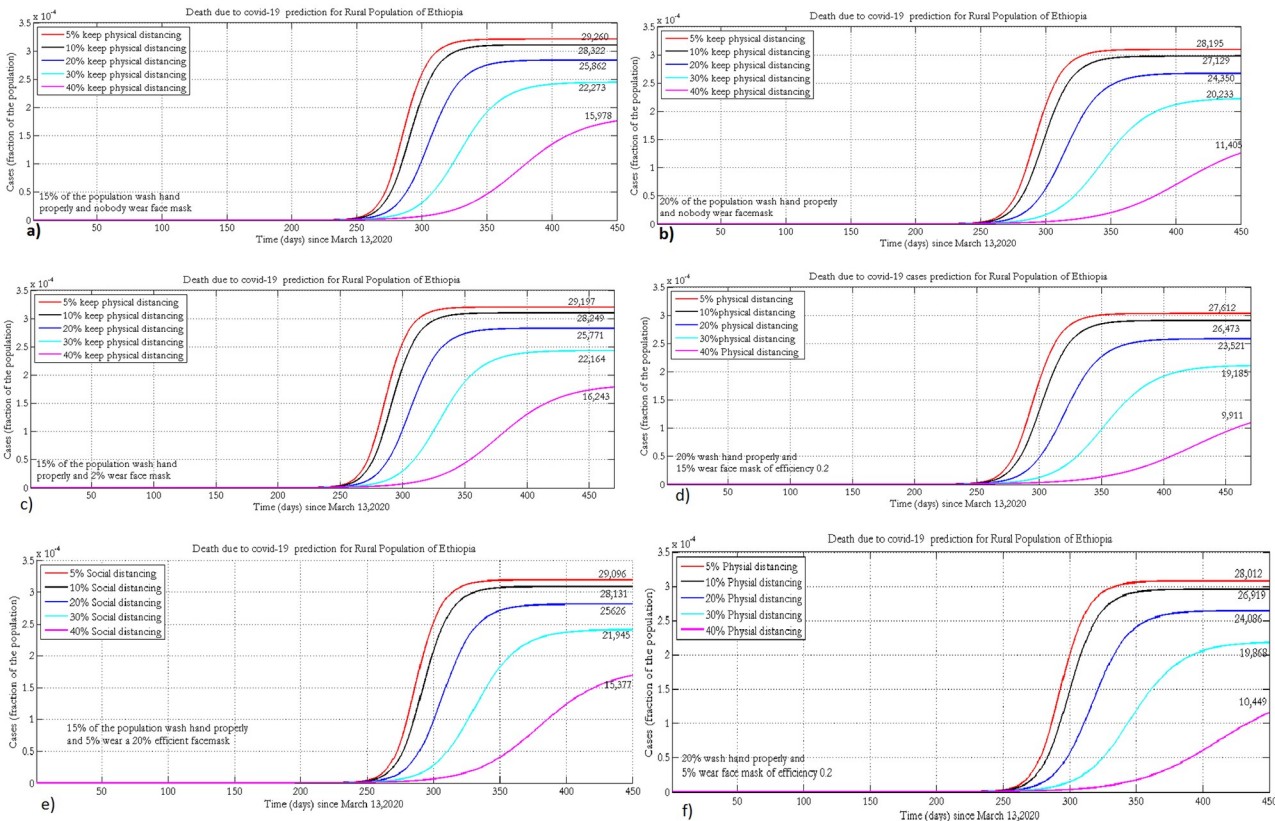

**Fig 9. Projected number of cumulative death due to COVID-19 in rural Ethiopia when 15% or 20% practice hand washing and 0% (a, b), 2% (c, d) or 5% (e, f) of the population wear face masks, with 20% efficacy.**

and social distancing and hand washing $SD - HW$, respectively, which may not necessarily be correlated.

Further, to evaluate the performance of the proposed model, we consider the actual reported cases in the month of October as a validation data set. Under moderate implementation of NPIs

**Table 6. Summary of predicted numbers of active COVID-19 and ICU cases (in thousands) under different adherence levels of mask and physical distancing in the urban population of Ethiopia.**

| Distancing | Percentage of population who wear face masks of efficiency 35%, Hygiene = 30% | | | | | | | | | | | |
|---|---|---|---|---|---|---|---|---|---|---|---|---|
| | Projected Number of all active Cases | | | | | | Projected number of active ICU cases | | | | | |
| | 5 | | 15 | | 25 | | 5 | | 15 | | 25 | |
| | Size | Period | Size | Period | Size | Period | Size | Period | Size | Period | Size | Period |
| 5 | 348 | Nov,2020 | 300 | Nov,2020 | 265 | Nov,2020 | 2.21 | Nov,2020 | 1.92 | Nov,2020 | 1.68 | Nov,2020 |
| 10 | 292 | Nov,2020 | 248 | Nov,2020 | 214 | Nov,2020 | 1.85 | Nov,2020 | 1.58 | Nov,2020 | 1.37 | Dec,2020 |
| 20 | 183 | Dec,2020 | 147 | Dec,2020 | 119 | Dec,2020 | 1.16 | Dec,2020 | 0.95 | Dec,2020 | 0.79 | Dec,2020 |
| 30 | 91 | Dec,2020 | 71 | Dec,2020 | 57 | Nov,2020 | 0.5 | Dec,2020 | 0.46 | Dec,2020 | 0.37 | Nov,2020 |
| 40 | 44 | Oct,2020 | 42 | Oct,2020 | 42 | Oct,2020 | 0.2 | Oct,2020 | 0.2 | Oct,2020 | 0.27 | Nov,2020 |
| | Percentage of population who wear-facemasks of efficiency 35%, Hygiene = 40% | | | | | | | | | | | |
| 5 | 199 | Dec,2020 | 165 | Dec,2020 | 137 | Dec,2020 | 1.26 | Dec,2020 | 1.04 | Dec,2020 | 0.8 | Dec,2020 |
| 10 | 156 | Dec,2020 | 126 | Dec,2020 | 103 | Dec,2020 | 0.9 | Dec,2020 | 0.7 | Dec,2020 | 0.6 | Dec,2020 |
| 20 | 82 | Dec,2020 | 65 | Nov,2020 | 57 | Nov,2020 | 0.5 | Dec,2020 | 0.4 | Nov,2020 | 0.3 | Nov,2020 |
| 30 | 45.4 | Oct,2020 | 44 | Oct,2020 | 44 | Oct,2020 | 0.2 | Oct,2020 | 0.2 | Oct,2020 | 0.2 | Oct,2020 |
| 40 | 45.2 | Oct,2020 | 44 | Oct,2020 | 44 | Oct,2020 | 0.2 | Oct,2020 | 0.2 | Oct,2020 | 0.2 | Oct,2020 |

**Table 7. Summary of predicted numbers of active COVID-19 and ICU cases (in thousands) under different adherence levels of mask wearing and physical distancing in the rural population of Ethiopia.**

| Distancing | Percentage of population who wear-facemasks of efficiency 20%, Hygiene = 15% | | | | | | | | | | | |
| --- | --- | --- | --- | --- | --- | --- | --- | --- | --- | --- | --- | --- |
| | Projected Number of all active Cases | | | | | | Projected number of active ICU cases | | | | | |
| | 0 | | 2 | | 5 | | 0 | | 2 | | 5 | |
| | Size | Period | Size | Period | Size | Period | Size | Period | Size | Period | Size | Period |
| 5 | 2736 | Dec,2020 | 2714 | Dec,2020 | 2680 | Dec,2020 | 11 | Dec,2020 | 11 | Dec,2020 | 11 | Dec,2020 |
| 10 | 2441 | Dec,2020 | 2420 | Dec,2020 | 2388 | Dec,2020 | 10 | Dec,2020 | 10 | Dec,2020 | 10 | Dec,2020 |
| 20 | 1829 | Jan,2021 | 1809 | Jan,2021 | 1780 | Jan,2021 | 7 | Jan,2021 | 7 | Jan,2021 | 7 | Jan,2021 |
| 30 | 1199 | Feb,2021 | 1181 | Feb,2021 | 1155 | Feb,2021 | 5 | Feb,2021 | 5 | Feb,2021 | 5 | Feb,2021 |
| 40 | 592 | Mar,2021 | 578 | Mar,2021 | 558 | Mar,2021 | 2 | Mar,2021 | 2 | Mar,2021 | 2 | Mar,2021 |
| | Hygiene improved from 15% to 20% | | | | | | | | | | | |
| 5 | 2404 | Dec,2020 | 2384 | Dec,2020 | 2353 | Dec,2020 | 10 | Dec,2020 | 10 | Dec,2020 | 9 | Dec,2020 |
| 10 | 2121 | Jan,2021 | 2100 | Jan,2021 | 2069 | Jan,2021 | 8 | Jan,2021 | 8 | Jan,2021 | 8 | Jan,2021 |
| 20 | 1533 | Jan,2021 | 1514 | Jan,2021 | 1485 | Jan,2021 | 6 | Jan,2021 | 6 | Jan,2021 | 6 | Jan,2021 |
| 30 | 942 | Feb,2021 | 926 | Feb,2021 | 902 | Feb,2021 | 3 | Feb,2021 | 3 | Feb,2021 | 3 | Feb,2021 |
| 40 | 399 | Mar,2021 | 387 | Mar,2021 | 370 | Mar,2021 | 1 | Mar,2021 | 1 | Mar,2021 | 1 | Mar,2021 |

(20% -social distancing, 5% -face mask coverage, and 25% hand washing), we compared projected cases of with data available from the Ministry of Health. The projected and observed cases are shown in Fig 11. The projected cases are very close to those observed, and there is a similar slope, indicating that our model is successfully capturing the dynamics of Covid-19 in Ethiopia. The projected number of cases (red line) by the proposed model are a bit higher than the actual cases (Fig 11). This may be due to under reported cases because of very low test coverage in Ethiopia. Further, the difference between the actual reported number of cases and projected cases ranges from 2053 to 6946 individuals which is an acceptable prediction error in a country with a large population (above 110 million in the case of Ethiopia).

## 4 Discussion

In this study, a modified SEIR model was developed to project the number of COVID-19 cases with different stages of the disease under the implementation of wearing face masks, hand washing and physical distancing with varying adherence levels in both urban and rural populations of Ethiopia. Unlike other mathematical modeling studies done in the context of Ethiopia

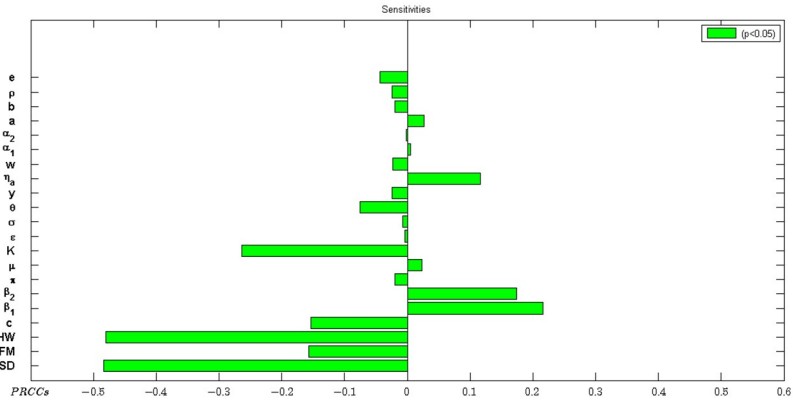

**Fig 10. Sensitivity of parameters with respect to $R_o$.**

**Table 8. Summary of different pairs of significant parameters by accounting for the false discovery rate.**

|  | SD | FM | HW | c | $\beta_1$ | $\varepsilon$ | $\theta$ | y | $\eta_a$ | w | e |
|---|---|---|---|---|---|---|---|---|---|---|---|
| SD |  | TRUE | FALSE | TRUE | TRUE | TRUE | TRUE | TRUE | TRUE | TRUE | TRUE |
| FM |  |  | TRUE | FALSE | TRUE | TRUE | TRUE | FALSE | TRUE | TRUE | TRUE |
| HW |  |  |  | TRUE | TRUE | TRUE | TRUE | TRUE | TRUE | TRUE | TRUE |
| c |  |  |  |  | TRUE | TRUE | TRUE | FALSE | TRUE | TRUE | TRUE |
| $\beta_1$ |  |  |  |  |  | TRUE | TRUE | TRUE | TRUE | TRUE | TRUE |
| $\varepsilon$ |  |  |  |  |  |  | TRUE | TRUE | TRUE | TRUE | FALSE |
| $\theta$ |  |  |  |  |  |  |  | TRUE | TRUE | TRUE | TRUE |
| y |  |  |  |  |  |  |  |  | TRUE | TRUE | TRUE |
| $\eta_a$ |  |  |  |  |  |  |  |  |  | TRUE | TRUE |
| w |  |  |  |  |  |  |  |  |  |  | TRUE |
| e |  |  |  |  |  |  |  |  |  |  |  |

[23, 24, 45], this study provides projected numbers of all active cases, symptomatic and asymptomatic cases and ICU cases at the peak time of the pandemic. The projected number of people in each stage of the disease at the peak period and the time of the peak in urban and rural areas of Ethiopia helps the government to choose and enforce better intervention mechanisms.

Our projection shows that, in line with [42, 46] improving the percentage of the population who wear face masks by 10% will protect around 100,000 more people from being infected by COVID-19 in the urban population. Regardless of face mask coverage and hygiene, the peak time of COVID-19 could be shifted to the future by two months if 40% of the population implements social distancing properly. Similar to our findings, the study by [23] demonstrated practicing physical distancing and wearing face masks delayed the peak time of the pandemic. [47] estimated the COVID-19 case burden for all African countries under different intervention scenarios (no intervention, moderate lock-down and hard lock-down) using a compartmental model. [48] studies the impacts of pharmaceutical and non-pharmaceutical interventions on COVID-19 in South Africa using a mathematical model and additionally, proper implementation of physical distancing greatly reduced the basic reproduction number.

In the urban population setting, if 20% of the population implements physical distancing and 30% adopt hygiene measures, and 25% wear face masks the peak time of the pandemic will happen in December, 2020 with 119,000 estimated cases (Table 6). During the peak time of the pandemic, except for some scenarios (30% physical distancing and above ≥15% face

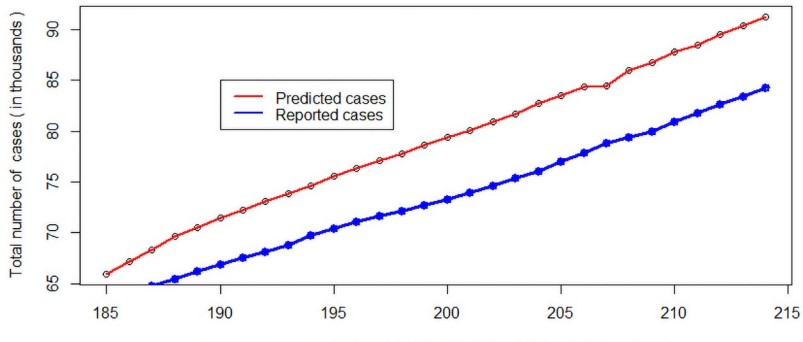

**Fig 11. Model validation.**

mask coverage), the projected number of ICU cases is above the capacity of the Ethiopian health system, an issue which therefore needs government attention (Table 6).

COVID-19 is a deadly virus that recent global efforts have led to a breakthrough in development of a number of effective vaccines being made available under the COVAX facility. Specific medical treatment is not yet available but supportive, therefore is highly recommended. Hence the government should focus on prevention of infection using NPI mechanisms. Low socioeconomic status and behavioral attitudes lead to closer interactions in congested households, hampering the successful implementation of social distancing measures. Lack or improper use of masks and use of low quality masks may exacerbate the spread of infection [49]. It is therefore imperative that the prudent use of NPIs is implemented in Ethiopia to contain the pandemic. NPIs used in combination are able to decrease cases and fatalities due to Covid-19. Separate implementation of each of the NPIs shortens the peak time of the pandemic and the number of new cases will increase by two-fold. We considered the place of residence being urban and rural as a factor since adherence to the recommended NPIs and the life style and living conditions, varies greatly between urban and rural populations of Ethiopia [50]. Similar to the urban population projection, projection of cases in the rural setting was done by modifying the model parameters in the context of rural population practices. Table 7 presents a summary of projected active COVID-19 cases and ICU cases in rural population of Ethiopia under 30 scenarios. The results shows that, at the peak of the pandemic, the number of projected cases in rural population will be higher than the urban population.

A better understanding of the impact of NPIs on COVID-19 transmission dynamics could help to control the spread of the disease. In future we suggest that extending the proposed approach by taking into account the age structure of the population and considering a stochastic modeling framework may lead to a more precise projection on the number of COVID-19 cases. Further, implementing the presented approach to other countries in the region will be helpful to provide insightful recommendations on the implementation of different NPIs to mitigate COVID-19 in the Horn of Africa.

## 5 Conclusion

The first COVID-19 case was detected on March 13, 2020 and since then the government of Ethiopia has applied different mitigation strategies. But the virus continues to spread. Since community transmission of the virus is established, temporary relaxation of any of the three non-pharmaceutical interventions could cause a rebound of the number of new cases, potentially leading to a collapse of the health care system. Our findings confirm, in the context of Ethiopia, that suppression of COVID-19 could be achieved by the combined implementation of three public health measures: wearing of face masks, social distancing and hygiene. The most effective public health measures are face mask wearing in urban populations and social distancing in rural populations. In addition, the successful enforcement of all these public health measures and guidelines by the government requires adherence by the public. Based on the projected results under different scenarios, concerned health stakeholders could recommend feasible and cost effective NPIs to policy makers for implementation.

## 6 Limitations

Studies on COVID-19 showed the distribution of cases strongly depended on age. We could not segregate our analysis into age-structured predictions or assess the impact of school closure on the pandemic due to lack of reliable resources. Further, the projection to the rural population setting is less reliable as the analysis has been constrained by different factors (i.e. the breakdown of case in rural settings was not known with precision, adherence to the

implemented NPIs vary greatly across different regions of the country, estimated number of protected individuals assumed 20%, and others). In addition to this, the testing rate of COVID-19 is very low in Ethiopia which may lead underestimation of the actual case number. This study did not account for under reporting of COVID-19 cases.

## Supporting information

**S1 File.**
(PDF)

**S1 Data.**
(ZIP)

## Acknowledgments

The research team acknowledges Semu M. Kassa (Botswana International University of Science and Technology, Botswana), and late Birhanu T. Ayele (Stellenbosch University, South Africa) for their valuable comments on the development of this work.

## Author Contributions

**Conceptualization:** Bedilu Alamirie Ejigu, Manalebish Debalike Asfaw.

**Data curation:** Bedilu Alamirie Ejigu, Manalebish Debalike Asfaw, Tilahun Abebaw.

**Formal analysis:** Bedilu Alamirie Ejigu, Manalebish Debalike Asfaw.

**Methodology:** Bedilu Alamirie Ejigu, Manalebish Debalike Asfaw.

**Supervision:** Lisa Cavalerie, Mark Nanyingi, Matthew Baylis.

**Validation:** Lisa Cavalerie, Matthew Baylis.

**Writing – original draft:** Bedilu Alamirie Ejigu, Manalebish Debalike Asfaw.

**Writing – review & editing:** Lisa Cavalerie, Tilahun Abebaw, Mark Nanyingi, Matthew Baylis.

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
