## [Decision Letter · Decision Letter 0]

24 Dec 2020

PONE-D-20-37247

Assessing the impact of non-pharmaceutical interventions (NPI) on the dynamics of COVID-19: A mathematical modelling study in the case of Ethiopia

PLOS ONE

Dear Dr. Bedilu Alamirie Ejigu,

Thank you for submitting your manuscript to PLOS ONE. After careful consideration, we feel that it has merit but does not fully meet PLOS ONE’s publication criteria as it currently stands. Therefore, we invite you to submit a revised version of the manuscript that addresses the points raised during the review process.

We look forward to receiving your revised manuscript.

Kind regards,

Yuka Kotozaki

Academic Editor

PLOS ONE

Journal Requirements:

2.) Thank you for stating the following financial disclosure:

'The funders had no role in study design, data collection and analysis, decision to publish, or preparation of the manuscript.'

3.) Please amend your authorship list in your manuscript file to include author Mark Nanyingi.

4.) Please ensure that you refer to Figure 1 in your text as, if accepted, production will need this reference to link the reader to the figure.

Reviewers' comments:

Reviewer's Responses to Questions

**Comments to the Author**

1. Is the manuscript technically sound, and do the data support the conclusions?

Reviewer #1: No

2. Has the statistical analysis been performed appropriately and rigorously? 

Reviewer #1: Yes

3. Have the authors made all data underlying the findings in their manuscript fully available?

Reviewer #1: No

4. Is the manuscript presented in an intelligible fashion and written in standard English?

Reviewer #1: Yes

5. Review Comments to the Author

Reviewer #1: This paper presents a mathematical model and analysis to investigate the impact of alternative non-pharmaceutical interventions (NPI) on the spread of COVID in Ethiopia. It builds upon a standard epidemiological model (SEIR) and adds components for symptomatic and asymptomatic infected individuals, and hospitalized individuals. It also adds a component for transmission from the environment. The model considers the impacts of NPI by essentially decreasing the transmission rates between model components. The model has 9 state variables covered by a standard system of ODEs with 22 parameters with values estimated from the limited data available or assumed. The model is used to project the dynamics of the state variables under several NPI scenarios separately for the urban and rural populations of Ethiopia.

The assessment of this paper would have been enhanced if the following issues had been addressed.

1. The introduction says "We believe that, due to the difference in the age-structure of the population, social interaction and life style in Ethiopia, mathematical models developed in other countries may not work to study the dynamics of disease in lower income settings." Given the tremendous number of COVID models already developed, this statement requires justification, particularly because the model they use has nothing at all about age-structure and the manner in which the differences in life style and social interactions are included in the model would be readily accounted for in standard SEIR models by simple modifications of the parameter values.

2. The model consists of two non-linear ODEs in a standard epidemiological format, linked to a linear system of ODEs for all the other 7 state variables. The system is therefore essentially equivalent in terms of long-term dynamics to a system of three equations and standard general theory of ODEs can be applied to show the resmdeified - the paper ults of Theorem 2.1 that is proved in the appendix, as well as the other two Theorems (indeed the authors refer to a previous paper for the proof if Theorem 2.2 and say nothing at all about Theorem 2.3 except state it. Thus the paper can be greatly reduced in length by referring to general ODE results.

3. There is no mention of evaluation nor is there any attempt to evaluate the model. With no criteria provided to state whether the model is appropriate, it s not clear that the projection they produce are at all meaningful. At the least, I would expect some parameterization be chosen and compared to the available data for a portion of the time series of at a and then use to project for the later part of the time series with some criteria applied to infer the model is reasonable. As it stands the implication from Table 4 is that the core dynamics of the infection was changing rapidly over the course of the year so that it is far from clear that the model assumptions fo constant parameter values is reasonable.

4. The paper is missing key descriptions that would be required to allow repeatability of the results stated. It is really not at all clear how the many parameters were estimated. For example the virus decay rate is stated as 1/4 with no source except "average" and many of the sources of the parameters are "assumed". Similarly the assumed differences between the urban and rural populations are assumed to be very specific values (27.4% and 7.8% of the urban and rural populations wash their hands) with little assessment that these could be tremendously off. There is no code listed or methods describe for the dynamic solutions of the systems of equations in the model -m presumably they are using some standard ODE solver but they do not say.

5. There is no discussion of data quality in the data sets on the disease progression, and in fact it isn't clear how these data sets were utilized in the model analysis. Given the variety of concerns about reporting inadequacies n many countries, some discussion of the implications of poor data should be included.

6. The main results in Fg 10 on the sensitivity analysis are completely obvious from the formula for R0 given that R0 has a factor of r4 which has symmetric negative dependence on SD, c FM and HW. Similarly the dependence on Beta1 enters because Beta1is a factor in the two terms of the R0.

7. Major portion of the manuscript is in Figures 2-9 which illustrate the dynamics of the model for the rural or urban population under a limited set of scenarios for NPI but there is no justification given for the scenarios chosen so the reader has no way of asserting that these are scenarios which are indeed more meaningful than the many other ones that are possible.

8. There are many, many places where the grammar and sentence structure needs to be modified. Similarly many of the references are not in appropriate style. Figure 1 is confusing - some dashed lines have arrows and others do not and it isn't clear why some are dashed and some are not based on the equations - the Env equation has a dashed line from S for some reason. The notation for the state variable I Hm is confusing since it seems to be two variables.

9. The authors assume that there is no movement of individuals at all between the urban and rural populations. This justifies their efforts to look at the NPI in the two separately. However this assumption needs to be justified.

6. PLOS authors have the option to publish the peer review history of their article (what does this mean?). If published, this will include your full peer review and any attached files.

Reviewer #1: No

---

## [Author Response · Author response to Decision Letter 0]

16 Feb 2021

Editor comments.

1.) Please ensure that your manuscript meets PLOS ONE's style requirements, including those for file naming.

- Addressed.

2.) Thank you for stating the following financial disclosure:

'The funders had no role in study design, data collection, and analysis, decision to publish, or preparation of the manuscript.'

1. Please clarify the sources of funding (financial or material support) for your study. List the grants or organizations that supported your study, including funding received from your institution.

2. State what role the funders took in the study. If the funders had no role in your study, please state: “The funders had no role in study design, data collection, and analysis, decision to publish, or preparation of the manuscript.”

3. If any authors received a salary from any of your funders, please state which authors and which funders.

4. If you did not receive any funding for this study, please state: “The authors received no specific funding for this work.”

- Addressed. In the revised cover letter, the following statement included.

 “This work was supported by Addis Ababa University, Ethiopia, and as a sandpit project of the One Health Regional Network for the Horn of Africa (HORN) project. The HORN project is funded by UK Research and Innovation (UKRI) and the Global Challenges Research Fund (GCRF) from the Growing Research Capability call awarded to the University of Liverpool, UK (project number BB/P027954/1). Funders had no role in the design, analysis, and interpretation of the results. Some of the Authors (Bedilu A. Ejigu, Manalebish D. Asfaw, Lisa Cavalerie, and Tilahun Abebaw) received salary.”

3.) Please amend your authorship list in your manuscript file to include author Mark Nanyingi.

- Corrected.

4.) Please ensure that you refer to Figure 1 in your text as, if accepted, production will need this reference to link the reader to the figure.

- In the revised version, Figure 1 cross-referenced.

Reviewers' comments:

Reviewer #1: 

This paper presents a mathematical model and analysis to investigate the impact of alternative non-pharmaceutical interventions (NPI) on the spread of COVID in Ethiopia. It builds upon a standard epidemiological model (SEIR) and adds components for symptomatic and asymptomatic infected individuals, and hospitalized individuals. It also adds a component for transmission from the environment. The model considers the impacts of NPI by essentially decreasing the transmission rates between model components. The model has 9 state variables covered by a standard system of ODEs with 22 parameters with values estimated from the limited data available or assumed. The model is used to project the dynamics of the state variables under several NPI scenarios separately for the urban and rural populations of Ethiopia.

The assessment of this paper would have been enhanced if the following issues had been addressed.

1. The introduction says "We believe that, due to the difference in the age structure of the population, social interaction and lifestyle in Ethiopia, mathematical models developed in other countries may not work to study the dynamics of disease in lower-income settings." Given the tremendous number of COVID models already developed, this statement requires justification, particularly because the model they use has nothing at all about age-structure and the manner in which the differences in lifestyle and social interactions are included in the model would be readily accounted for in standard SEIR models by simple modifications of the parameter values.

- We thankfully accept the comment about the age-structure. Initially, we assumed we would get data stratified by age. In the meantime, it was difficult to find such data, and in the revised version we amended the text accordingly. Regarding the lifestyle, urban and rural population differs in a number of ways. Such as, in rural areas, people live in a more scattered way than the urban, and the way they greet when they meet each other is also different (i.e. in rural areas there is no custom of handshaking). Further, the attention they give to hygiene is another source of differences considered in the paper. 

2. The model consists of two non-linear ODEs in a standard epidemiological format, linked to a linear system of ODEs for all the other 7 state variables. The system is therefore essentially equivalent in terms of long-term dynamics to a system of three equations and standard general theory of ODEs can be applied to show the resmdeified - the paper ults of Theorem 2.1 that is proved in the appendix, as well as the other two Theorems (indeed the authors refer to a previous paper for the proof if Theorem 2.2 and say nothing at all about Theorem 2.3 except state it. Thus the paper can be greatly reduced in length by referring to general ODE results.

- We thankfully accept the comment. Details on the mathematical model system equilibria, stability of the system, sensitivity analyses, and well-posedness of the proposed mathematical model were provided as supplementary material of the revised version.

3. There is no mention of evaluation nor is there any attempt to evaluate the model. With no criteria provided to state whether the model is appropriate, it’s not clear that the projection they produce are at all meaningful. At the least, I would expect some parameterization to be chosen and compared to the available data for a portion of the time series of at a and then use to project for the later part of the time series with some criteria applied to infer the model is reasonable. As it stands the implication from Table 4 is that the core dynamics of the infection was changing rapidly over the course of the year so that it is far from clear that the model assumptions of constant parameter values is reasonable.

- We thankfully accept the comment. Because of little surveillance testing for COVID-19 in Ethiopia, it is difficult to get suitable data for model validation. Our aim in this study was to project the course of the disease in Ethiopia at a point in time depending on different adherence levels of NPIs. Due to the limited availability of data in Ethiopia we have used values from the literature to parameterize our model. However to assess the importance of considered parameters in the considered model, we have performed a sensitivity analysis by computing partial rank correlation coefficients (PRCC). Further, to see whether one parameter depends on another, pair wise comparisons were carried out.

- The difference in Ro (presented in Table 4) over the course of the dynamics is due to the intervention enforced by the government. Our model is helpful in indicating trends for projected cases depending on different levels of NPIs implementation.

4. The paper is missing key descriptions that would be required to allow repeatability of the results stated. It is really not at all clear how many parameters were estimated. For example, the virus decay rate is stated as 1/4 with no source except "average" and many of the sources of the parameters are "assumed". Similarly, the assumed differences between the urban and rural populations are assumed to be very specific values (27.4% and 7.8% of the urban and rural populations wash their hands) with little assessment that these could be tremendously off. There is no code listed or methods described for the dynamic solutions of the systems of equations in the model -m presumably they are using some standard ODE solver but they do not say.

- We thankfully accept the comment and in the revised version, sources were explicitly mentioned (Table 1, last column), and additional references were included in the reference list. 

- A number of factors (temperatures, relative humidity, and UV Index) influence the COVID-19 virus decay rate and we consider the average. In the revised manuscript references were given.

- The assumed difference between the urban and rural populations (27.4% and 7.8% of the urban and rural populations wash their hands) is evidence based as revealed by the 2016 Ethiopian demographic and health survey. Reference sources included in the revised version of the manuscript.

- For repeatability of the results, matlab codes used to generate the results in the manuscript and now included in the supplementary material.

5. There is no discussion of data quality in the data sets on the disease progression, and in fact it isn't clear how these data sets were utilized in the model analysis. Given the variety of concerns about reporting inadequacies in many countries, some discussion of the implications of poor data should be included.

- We have used publicly available data (i.e. daily number of new cases, deaths) and some variables (i.e. daily number of ICU cases) extracted from ministry of health and Ethiopian public health institute reports. The testing rate is indeed quite low in Ethiopia compared to other countries, however given the fact that the number of cases is only uses as indicative trend prior to our projection, this has little consequences on our model projections

- We mentioned the impact of poor data on the “Limitation” subsection of the manuscript. The testing rate of COVID-19 is very low in Ethiopia which may lead the number reported of cases is underestimated.

6. The main results in Fg 10 on the sensitivity analysis are completely obvious from the formula for R0 given that R0 has a factor of r4 which has symmetric negative dependence on SD, c FM and HW. Similarly, the dependence on Beta1 enters because Beta1is a factor in the two terms of the R0.

- Yes that is true. But, for the general audience, the equation for Ro is not obvious. Further, the sensitivity analysis graph is a better visualization guide for future efforts to better describe the impact of hygiene, social distancing and wearing facemask. In addition to this, this section also tries to investigate whether one parameter depends on another using a pair wise comparison test. 

7. Major portion of the manuscript is in Figures 2-9 which illustrate the dynamics of the model for the rural or urban population under a limited set of scenarios for NPI but there is no justification given for the scenarios chosen so the reader has no way of asserting that these are scenarios which are indeed more meaningful than the many other ones that are possible.

- We thankfully accept the comment, and in the revised manuscript, the reason for considering those scenarios was given in the revised method Section 2.3 as follow. “These values were selected based on expert opinion of how the dynamics may look under the implementation of NPIs with different adherence levels in the urban and rural populations. Further, the considered NPI’s coverage is supported by phone-based survey results. (Baye 2020, Kebede et al 2020).”

8. There are many, many places where the grammar and sentence structure needs to be modified. Similarly, many of the references are not inappropriate style. Figure 1 is confusing - some dashed lines have arrows and others do not and it isn't clear why some are dashed and some are not based on the equations - the Env equation has a dashed line from S for some reason. The notation for the state variable IHm is confusing since it seems to be two variable

- Flow diagram of the mathematical model showing the transition of individuals in different compartments based on infectious status (solid lines). The model also includes the environment, which receives infection from asymptomatic and symptomatic individuals and from which infection can pass to susceptible individuals (dashed lines), at a rate dependent on the force of infection.

- To avoid confusing, the state variable IHm changed to Hm.

9. The authors assume that there is no movement of individuals at all between the urban and rural populations. This justifies their efforts to look at the NPI in the two separately. However, this assumption needs to be justified.

- Your concern in this respect is valid. We draw this assumption due to the fact that there was very low road accessibility to connect the urban and rural population of Ethiopia (World-Bank (2016). Measuring Rural Access).

---

## [Decision Letter · Decision Letter 1]

25 Mar 2021

PONE-D-20-37247R1

Assessing the impact of non-pharmaceutical interventions (NPI) on the dynamics of COVID-19: A mathematical modelling study in the case of Ethiopia

PLOS ONE

Dear Dr. Bedilu Alamirie Ejigu,

Thank you for submitting your manuscript to PLOS ONE. After careful consideration, we feel that it has merit but does not fully meet PLOS ONE’s publication criteria as it currently stands. Therefore, we invite you to submit a revised version of the manuscript that addresses the points raised during the review process.

We look forward to receiving your revised manuscript.

Kind regards,

Yuka Kotozaki

Academic Editor

PLOS ONE

Reviewers' comments:

Reviewer's Responses to Questions

**Comments to the Author**

1. If the authors have adequately addressed your comments raised in a previous round of review and you feel that this manuscript is now acceptable for publication, you may indicate that here to bypass the “Comments to the Author” section, enter your conflict of interest statement in the “Confidential to Editor” section, and submit your "Accept" recommendation.

Reviewer #1: (No Response)

2. Is the manuscript technically sound, and do the data support the conclusions?

Reviewer #1: Partly

3. Has the statistical analysis been performed appropriately and rigorously? 

Reviewer #1: No

4. Have the authors made all data underlying the findings in their manuscript fully available?

Reviewer #1: No

5. Is the manuscript presented in an intelligible fashion and written in standard English?

Reviewer #1: No

6. Review Comments to the Author

Reviewer #1: The authors have added a supplementary information component to this paper that improves the clarity of the mathematical results.

The manuscript does not state what numerical method they are using in the Matlab code - it appears they are using the Euler method and some justification why a more standard ODE solver within Matlab is not provided. Some justification for the numerical stability of the Euler method they implement is needed since there are many reasons why this method is replaced by others. Some of the rate parameters in the model are one or two orders of magnitude different from other rate parameters, which implies the potential for stiffness, which is one of the conditions under which the Euler method is unstable numerically.

In response to comments regarding model evaluation (review comment #3), the authors still make no attempt to compare the model results with data. They state that the sensitivity analysis provides insight into the importance of model parameters, which it does, but sensitivity analysis provides no evidence in and of itself that the model assumptions and results appropriately describe the dynamics of the system so as to be useful.

In response to review comment #3 the authors state that the results in Table 4 arose from different "interventions enforced by the government" over the time period modeled. However there is no statement about government timing of interventions in the data section. It appears Table 4 gives model results assumed to be from the interventions and adherence levels given in Table 2, but there is no mention of the timing of these interventions. So if the interventions start at the beginning of the time period and remain constant, why are the R0s changing so much over time in the model results in Table 4?

The references in the revision are still not formatted consistently.

This article has still not been copyedited and since PLOS does not copy edit manuscripts this manuscript still Neds to be corrected in many places for grammatical reasons.

Figure 1 is still unclear - there are still dashed lines with no arrows and one dashed line with an arrow - what do the dashed lines represent? The arrows are still left out.

Table 1 - the values are a mixture of fraction and decimal notation - these should be uniform

Supplementary information Comments:

1. In Theorem 0.1 there is a mention of Theorem 2.1 after the Theorem statement - what is this in reference to?

2. Sensitivity analysis section states :

The partial derivative of the threshold value Ro with

respect to the input parameters were computed by varying the parameters around

normal values.

What are "normal values"

3. In the paragraph after eq (11) there is a [?] ref

4. The Matlab codes provided have no overall description, nor do they appear to include the all calculations discussed in the paper - for example there does not appear to be any sensitivity analysis included in the Matlab codes

4. The references in the supplement are not in standard uniform style

7. PLOS authors have the option to publish the peer review history of their article (what does this mean?). If published, this will include your full peer review and any attached files.

Reviewer #1: No

---

## [Author Response · Author response to Decision Letter 1]

15 May 2021

Review Comments to the Author

Reviewer #1: The authors have added a supplementary information component to this paper that improves the clarity of the mathematical results.

The manuscript does not state what numerical method they are using in the Matlab code - it appears they are using the Euler method and some justification why a more standard ODE solver within Matlab is not provided. Some justification for the numerical stability of the Euler method they implement is needed since there are many reasons why this method is replaced by others. Some of the rate parameters in the model are one or two orders of magnitude different from other rate parameters, which implies the potential for stiffness, which is one of the conditions under which the Euler method is unstable numerically.

- There are several methods for solving ODEs. We prefer to use Euler because Euler method is used in solving differential equations with known initial conditions. Further this method also used to approximate the behavior of the equations involved in a certain range. To increase the accuracy of the method we used a time step of 0.01.

 In response to comments regarding model evaluation (review comment #3), the authors still make no attempt to compare the model results with data. They state that the sensitivity analysis provides insight into the importance of model parameters, which it does, but sensitivity analysis provides no evidence in and of itself that the model assumptions and results appropriately describe the dynamics of the system so as to be useful.

- In the revised version of the manuscript, we did the model evaluation using the actual data in the month of October. The following figure shows how the predicted number of cases (red line) looks as compared with the actual reported case by ministry of health (blue line). The result reveals that the predicted number of cases by the proposed model are a bit higher than the actual cases. This may be due to under reported cases because of very low test coverage in Ethiopia. Further, the difference between the actual reported number of cases and predicted cases ranges from 2053 to 6946 individuals which is acceptable prediction in error in a country with a large population (above 110 million in the case of Ethiopia). The two lines have similar slopes, and that the numbers of cases are similar given the margins of error and likely under-detection.

- Under moderate implementation of NPIs (20% -social distancing, 5%-face mask coverage, and 25% hand wash), we compared predicted cases of with data available from the ministry of health. The predicted and observed cases are shown in Fig 11. The predicted cases are very close to those observed, and there is a similar slope, indicating that our model is successfully capturing the dynamics of Covid-19 in Ethiopia. 

In response to review comment #3 the authors state that the results in Table 4 arose from different "interventions enforced by the government" over the time period modeled. However there is no statement about government timing of interventions in the data section. It appears Table 4 gives model results assumed to be from the interventions and adherence levels given in Table 2, but there is no mention of the timing of these interventions. So if the interventions start at the beginning of the time period and remain constant, why are the R0s changing so much over time in the model results in Table 4?

- Yes, the interventions start at the beginning of the time period and remain constant, while the adherence level in the population varies over time. For instance, in the week after the holidays of Easter and Ramadan (61-90 days since the onset of COVID in Ethiopia), there was increased travelling and intense interaction in the population. Moreover only schools remain closed, and social distancing in public transport and market places wasn’t adhered to. The government was able to support and enforce wearing of face masks in public but, due to socioeconomic constraints, was less able to enforce adherence to social distancing and hand washing.

- The MATLAB code calculates the new Ro considering time-varying levels of adherence to the NPIs. The MATLAB code included in the supplement. 

The references in the revision are still not formatted consistently.

 - Addressed.

This article has still not been copyedited and since PLOS does not copy edit manuscripts this manuscript still needs to be corrected in many places for grammatical reasons.

- Addressed. The revised manuscript edited by native speaker. 

Figure 1 is still unclear - there are still dashed lines with no arrows and one dashed line with an arrow - what do the dashed lines represent? The arrows are still left out.

- In Figure 1 we have two main compartments: i) Human population and ii) Environment. To delineate the human to human (H2H) interaction and progression from different states (SEIR) we used solid lines. The interaction of these human compartments with the environment are indicated by broken lines. The dashed lines represent potential shedding of the virus to the environment by individuals in the asymptomatic and symptomatic compartments, the susceptible population may be infected from the contaminated environment. Such schematic representations has been used in modeling infectious diseases involving contaminated environment such as malaria and cholera.

Table 1 - the values are a mixture of fraction and decimal notation - these should be uniform.

- Addressed 

 Supplementary information Comments

1. In Theorem 0.1 there is a mention of Theorem 2.1 after the Theorem statement - what is this in reference to?

- Thanks for pointing out this. It was a typo error and corrected in the revised version.

2. Sensitivity analysis section states: The partial derivative of the threshold value Ro with respect to the input parameters were computed by varying the parameters around normal values.What are "normal values".

- These values were obtained from the table of parameters the sensitivity analysis studies what would happen if we perturb the values within a certain interval.

3. In the paragraph after eq (11) there is a [?] ref

- A reference has been inserted.

4. The Matlab codes provided have no overall description, nor do they appear to include the all calculations discussed in the paper - for example there does not appear to be any sensitivity analysis included in the Matlab codes

- In the revised manuscript, a brief description of the Matlab codes is included. The sensitivity analysis code has also been included in the attached Matlab code.

5. The references in the supplement are not in standard uniform style.

All references have been updated using a uniform bibliographic format.

---

## [Decision Letter · Decision Letter 2]

1 Jun 2021

PONE-D-20-37247R2

Assessing the impact of non-pharmaceutical interventions (NPI) on the dynamics of COVID-19: A mathematical modelling study in the case of Ethiopia

PLOS ONE

Dear Dr. Bedilu Alamirie Ejigu,

Thank you for submitting your manuscript to PLOS ONE. After careful consideration, we feel that it has merit but does not fully meet PLOS ONE’s publication criteria as it currently stands. Therefore, we invite you to submit a revised version of the manuscript that addresses the points raised during the review process.

We look forward to receiving your revised manuscript.

Kind regards,

Yuka Kotozaki

Academic Editor

PLOS ONE

Reviewers' comments:

Reviewer's Responses to Questions

**Comments to the Author**

1. If the authors have adequately addressed your comments raised in a previous round of review and you feel that this manuscript is now acceptable for publication, you may indicate that here to bypass the “Comments to the Author” section, enter your conflict of interest statement in the “Confidential to Editor” section, and submit your "Accept" recommendation.

Reviewer #1: (No Response)

2. Is the manuscript technically sound, and do the data support the conclusions?

Reviewer #1: Partly

3. Has the statistical analysis been performed appropriately and rigorously? 

Reviewer #1: Yes

4. Have the authors made all data underlying the findings in their manuscript fully available?

Reviewer #1: Yes

5. Is the manuscript presented in an intelligible fashion and written in standard English?

Reviewer #1: No

6. Review Comments to the Author

Reviewer #1: This manuscript has been improved. It has ben copyedited, with many changes made. Most of the points raised by the previous review have been addressed, at least in part, in the manuscript. A few additional comments:

1. Although it has been copyedited, there are still places with grammatical errors, so another round of editing would be beneficials. A few places this reviewer saw that had errors are included below but another reading would likely catch other grammatical errors.

In the Abstract:

"Ethiopia under the COVAX facility began vaccinating high risk populations but due to vaccine supply shortages and the absence an effective treatment,"

Should be

"Ethiopia under the COVAX facility began vaccinating high risk populations but due to vaccine supply shortages and the absence of an effective treatment,"

In the second paragraph of the Introduction:

"By the mid of November, 2020 the cases had surpassed 100,000."

Should be

"By the middle of November, 2020 the cases had surpassed 100,000."

Figure 1 legend:

"The dashed lines represent potential shedding of the virus to the environment by individuals in the asymptomatic and symptomatic compartments, the susceptible population may be infected from the contaminated environment."

Should be:

"The dashed lines represent the potential shedding of the virus to the environment by individuals in the asymptomatic and symptomatic compartments as well as possible infection of the susceptible population from the contaminated environment."

2. Although the response says "All references have been updated using a uniform bibliographic format", this is far from correct. There are many references that remain incomplete. These should all be changed before publication. Examples of incomplete references from just the first 15 include:

1. WHO. Novel Coronavirus (2019-nCoV) situation reports. online. 2020;.

7. Lai S, Ruktanonchai NW, Zhou L, Prosper O, Luo W, Floyd JR. Effect of

non-pharmaceutical interventions to contain COVID-19 in China. Nature. 2020;.

8. Flaxman S, Mishra S, Gandy A, Unwin HJT, Mellan TA, Coupland H, et al.

Estimating the number of infections and the impact of non-pharmaceutical

interventions on COVID-19 in 11 European countries. Imperial College

COVID-19 Response Team. 2020;.

9. Ferguson NM, Laydon D, Nedjati-Gilani G, Imai N. Impact of

non-pharmaceutical interventions (NPIs) to reduce COVID-19 mortality and

healthcare demand,. Imperial College COVID-19 Response Team. 2020;.

10. Kassa SM, Njagarah HJB, Terefe YA. Analysis of the mitigation strategies for

COVID-19: from a mathematical modelling perspective. preprint. 2020;.

11. Anderson RM, May RM. Population biology of infectious diseases: Part I.. vol.

280. Nature; 1979.

12. Prem K, Liu Y, Russell TW, Kucharski AJ, Eggo RM, Davies N, et al. The effect

of control strategies to reduce Physical mixing on outcomes of the COVID-19

epidemic in Wuhan, China: a modelling study. Lancet Public Health. 2020;.

13. Li Q, Guan X, P Wu, Wang X, Zhou L, Tong Y, et al. Early transmission

dynamics in Wuhan, China, of novel coronavirus-infected pneumonia. N Engl J

Med. 2020;.

14. Walker PGT, Whittaker C, Wason OJ, Baguelin M, Winskill P, A H, et al. The

Global impact of COVID-19 and strategies for mitigation and suppression.

Science 369. 2020;.

15. Ivorra B, Ferrández MR, Vela-Pérez M, Ramos AM. Mathematical modeling of

the spread of the coronavirus disease 2019 (COVID-19) considering its particular characteristics. The case of China.; 2020. preprint.

7. PLOS authors have the option to publish the peer review history of their article (what does this mean?). If published, this will include your full peer review and any attached files.

Reviewer #1: No

---

## [Author Response · Author response to Decision Letter 2]

2 Jul 2021

Review Comments to the Author 

Reviewer #1: This manuscript has been improved. It has been copyedited, with many changes made. Most of the points raised by the previous review have been addressed, at least in part, in the manuscript. A few additional comments: 1. Although it has been copyedited, there are still places with grammatical errors, so another round of editing would be beneficials. A few places this reviewer saw that had errors are included below but another reading would likely catch other grammatical errors.

- Thanks for going through the details and providing constructive comments which greatly improves the quality of the paper. A second-round grammatical editing was done by the native speaker co-authors.

2. Although the response says "All references have been updated using a uniform bibliographic format", this is far from correct. There are many references that remain incomplete. These should all be changed before publication. Examples of incomplete references from just the first 15 include: 

 - Thanks for pointing out this. Your concern is correct and corrected in the revised version.

---

## [Decision Letter · Decision Letter 3]

29 Oct 2021

Assessing the impact of non-pharmaceutical interventions (NPI) on the dynamics of COVID-19: A mathematical modelling study in the case of Ethiopia

PONE-D-20-37247R3

Dear Dr. Ejigu,

We’re pleased to inform you that your manuscript has been judged scientifically suitable for publication and will be formally accepted for publication once it meets all outstanding technical requirements.

Kind regards,

Carla Pegoraro

Division Editor

PLOS ONE

Additional Division Editor Comments:

Please note that a new reviewer was invited by the previous Academic Editor and that they have made some minor suggestions along with recommending acceptance of your submission. If you are able to improve the quality of the Figures and potentially update the reference list during the final technical checks and production phase please do so. These are not a requirement for the Accept decision now been issued. Please accept my sincerest apologies for the delay in reaching this final decision.

Reviewers' comments:

Reviewer's Responses to Questions

**Comments to the Author**

1. If the authors have adequately addressed your comments raised in a previous round of review and you feel that this manuscript is now acceptable for publication, you may indicate that here to bypass the “Comments to the Author” section, enter your conflict of interest statement in the “Confidential to Editor” section, and submit your "Accept" recommendation.

Reviewer #2: All comments have been addressed

2. Is the manuscript technically sound, and do the data support the conclusions?

Reviewer #2: Yes

3. Has the statistical analysis been performed appropriately and rigorously? 

Reviewer #2: No

4. Have the authors made all data underlying the findings in their manuscript fully available?

Reviewer #2: (No Response)

5. Is the manuscript presented in an intelligible fashion and written in standard English?

Reviewer #2: Yes

6. Review Comments to the Author

Reviewer #2: (a) The quality of all the presented figures should be improved.

(b) More recent references regarding the Mathematical Modelling of COVID-19 should

be included.

(c) A brief description of determining the basic reproduction number R0 is needed.

7. PLOS authors have the option to publish the peer review history of their article (what does this mean?). If published, this will include your full peer review and any attached files.

Reviewer #2: No

---

## [Editor Report · Acceptance letter]

4 Nov 2021

PONE-D-20-37247R3 

Assessing the impact of non-pharmaceutical interventions (NPI) on the dynamics of COVID-19: a mathematical modelling study of the case of Ethiopia  

Dear Dr. Ejigu:

I'm pleased to inform you that your manuscript has been deemed suitable for publication in PLOS ONE. Congratulations! Your manuscript is now with our production department. 

Kind regards, 

on behalf of

Dr Carla Pegoraro 

Staff Editor

PLOS ONE